# Buildup of a highly twisted magnetic flux rope during a solar eruption

Wensi Wang[1], Rui Liu [1,2], Yuming Wang [1,3], Qiang Hu[4], Chenglong Shen[1,3], Chaowei Jiang [5,6] & Chunming Zhu[7]

The magnetic flux rope is among the most fundamental magnetic configurations in plasma. Although its presence after solar eruptions has been verified by spacecraft measurements near Earth, its formation on the Sun remains elusive, yet is critical to understanding a broad spectrum of phenomena. Here we study the dynamic formation of a magnetic flux rope during a classic two-ribbon flare. Its feet are identified unambiguously with conjugate coronal dimmings completely enclosed by irregular bright rings, which originate and expand outward from the far ends of flare ribbons. The expansion is associated with the rapid ribbon separation during the flare main phase. Counting magnetic flux through the feet and the ribbon-swept area reveals that the rope's core is more twisted than its average of four turns. It propagates to the Earth as a typical magnetic cloud possessing a similar twist profile obtained by the Grad-Shafranov reconstruction of its three dimensional structure.

[1] CAS Key Laboratory of Geospace Environment, Department of Geophysics and Planetary Sciences, University of Science and Technology of China, 230026 Hefei, China. [2] Collaborative Innovation Center of Astronautical Science and Technology, 230026 Hefei, China. [3] Synergetic Innovation Center of Quantum Information & Quantum Physics, University of Science and Technology of China, 230026 Hefei, China. [4] Center for Space Plasma and Aeronomic Research, The University of Alabama in Huntsville, Huntsville, AL 35899, USA. [5] Institute of Space Science and Applied Technology, Harbin Institute of Technology, 518055 Shenzhen, China. [6] SIGMA Weather Group, State Key Laboratory for Space Weather, Center for Space Science and Applied Research, Chinese Academy of Sciences, 100190 Beijing, China. [7] Department of Physics, Montana State University, Bozeman, MT 59717, USA. Correspondence and requests for materials should be addressed to R.L. (email: rliu@ustc.edu.cn)

A magnetic flux rope (MFR) consists of helical magnetic field lines collectively winding around a common axis[1]. To understand such a coherent structure is an important topic in many astrophysical, space, and laboratory contexts involving magnetized plasma[2]. In solar and heliospheric physics, the MFR is considered from both theoretical and observational perspectives a fundamental and key structure in solar eruptions[3, 4], which are manifested as diversely as coronal mass ejections (CMEs), solar flares, and prominence eruptions, but may be governed by similar physical mechanisms. These phenomena are the dominant contributor to adverse Space Weather at Earth and a laboratory for understanding the activity of more remote astronomical objects.

While it is a consensus that the free energy powering solar eruptions is stored in stressed (twisted or sheared) magnetic fields in the corona, the key parameters leading up to an eruption are still not understood; among them the nature of the preeruptive configuration has been under intense debate. Relevant to the debate are two prominent classes of CME/flare models that have been developed over the years. In the first, including the standard picture of solar flares, an MFR is present prior to the eruption[5–8]. This pre-existent MFR may be forced by magnetic buoyancy to emerge through the solar surface into the corona[9, 10], or form in the low corona by slow magnetic reconnection in a sheared magnetic arcade[11], which is driven by the gradual evolution of the magnetic field in the photospheric boundary[12]. In the second, the initial state typically contains sheared arcades and a new MFR forms via magnetic reconnection during the course of the eruption[13–15]. Magnetic reconnections during flares could add significant amount of magnetic fluxes into MFRs, as implied by a statistical comparison between the flux budget of interplanetary magnetic clouds (MCs) and the reconnection flux in their source regions[16, 17]. These reconnections are mapped to flare ribbons on the surface via field-aligned energy transport from the reconnection sites.

MFRs are clearly present after solar eruptions, as evidenced, in particular, by MCs detected at 1 AU (in situ), which possess a stronger, smoothly rotating magnetic field and a lower ion temperature than the ambient solar wind[18, 19], yet their formation back on the Sun remains elusive, mainly due to the insurmountable difficulty of measuring the coronal magnetic field[20]. The prominence-cavity system[21] and the sigmoidal hot emission (termed 'sigmoid'[22–24]) in soft X-rays (SXRs) or extreme ultraviolet (EUV)) are among the most trusted indications of MFRs on the Sun. However, the interpretation of these coronal features and their formation, the latter of which has been rare[25, 26], suffers inevitably from projection effects of these inherently three dimensional (3D) structures and the line-of-sight confusion in the optically thin corona, and hence has seldom been unambiguous.

Post-eruptive coronal dimming is naturally suggested to map the feet of eruptive MFRs, along which mass drainage into interplanetary space could take place[16, 27]. Some analytical models[7, 28, 29] demonstrate that an MFR is wrapped around by a thin volume of strong magnetic field distortion, known as quasi-separatrix layers (QSLs[30]). The photospheric footprints of the QSLs display two J-shaped ribbons, with the MFR anchored within the hooked parts. The hooks would close onto themselves if the MFR is highly twisted (three turns in ref. [28] and two turns in ref. [29]). This has not yet been verified by observation, whereas the 'open' double-J morphology is suggested to indicate a twist of no more than one turn[31]. It is generally believed that a pre-existent MFR should not possess a twist more than the threshold (~1.25 turns) of the helical kink instability[1]. In contrast, highly twisted MCs (up to six turns per AU) are indeed detected by in situ spacecraft measurements[17, 32]. It remains an open question how the high twist is produced.

Here we present observations of a highly twisted MFR dynamically formed during an eruptive flare. The associated CME arrived at Earth three days later as a typical interplanetary MC. The MFR's formation process and its twist distribution are deciphered from the morphology and dynamic evolution of flare ribbons, which is not subject to projection effects nor line-of-sight confusion, and further corroborated by in situ diagnostics of the resulting MC.

## Results

**Overview of the 4 November 2015 eruptive event.** The eruption of interest occurred close to the solar disk center on 4 November 2015 in a decaying active region, NOAA 12,443 (Supplementary Fig. 1; Methods section). It produced a halo CME propagating at ~600 km s$^{-1}$ observed in white light by the Large Angle and Spectrometric Coronagraph Experiment (LASCO), and an M3.7-class long-duration flare observed in various UV/EUV passbands by the Atmospheric Imaging Assembly (AIA[33]) onboard the Solar Dynamics Observatory (SDO; Methods section). The flare, peaking in SXRs at 13:52 UT, was also observed in hard X-rays (HXRs) by the Reuven Ramaty High-Energy Solar Spectroscopic Imager (RHESSI[34]). At first sight, the flare displays two semi-parallel ribbons in the chromosphere at two sides of the major polarity inversion line (PIL) that separates opposite polarities in the active region (Fig. 1), typical of a classic two-ribbon flare, but closer inspection reveals a morphology that has not been noticed before, i.e., two closed, irregular rings are developed during the flare main phase, attached to the far ends of flare ribbons (Figs. 2, 3), which is distinct from the frequently reported double-J-shaped ribbons with two open hooks.

The CME arrived at 1 AU three days later passing through the Advanced Composition Explorer (ACE) and WIND spacecrafts (Fig. 4). The interplanetary counterpart of the solar eruption ('IP ejecta' hereafter) began with the arrival of an interplanetary shock at 17:30 UT on 6 November 2015 until an increase in plasma $\beta$ (the ratio between thermal and magnetic pressure) at about 18:00 UT on 9 November 2015. This time interval was co-temporal with enhanced Fe/O ratio (known as the first ionization potential effect) as compared to the ambient solar wind. A typical MC was observed as the core of the IP ejecta, characterized by enhanced magnetic field strength, a smooth rotation of the field vector from south to north, and low plasma $\beta$. Hence, the solar eruption indisputably ejected an MFR into interplanetary space, regardless of whether that MFR is preexisting or newly formed. Below we will explore in detail the evolution of the flare morphology and demonstrate that the MFR is in fact dynamically formed during the eruption.

**Dynamic MFR formation.** Prior to the flare, a dark, east–west oriented filament (interchangeable with 'prominence' in the literature) was situated along the northern segment of the PIL (Fig. 1a1–a3; labeled 'F'). The filament had experienced a few episodes of moderate activities, but its feet (marked by boxes in Fig. 1a1, a2, e1) were relatively fixed (Supplementary Note 1). At about 13:22 UT, the filament was disturbed and heated, displaying intermixed dark and bright material (Fig. 1b2, b3; Supplementary Movie 1). This gives an impression of dark filament material threaded by highly sheared coronal loops (Fig. 1c3). These sheared loops emitted only in AIA 131 and 94 Å but not in any other passbands (Supplementary Movie 2), suggesting that they were heated to multi MK (Methods section). Meanwhile two flare ribbons parallel to the PIL started to form on the chromosphere, visible in AIA 1600 Å (Fig. 1c1). The flare ribbon located in the positive polarity area (labeled 'R+') mainly extended eastward while that in the negative polarity area (labeled

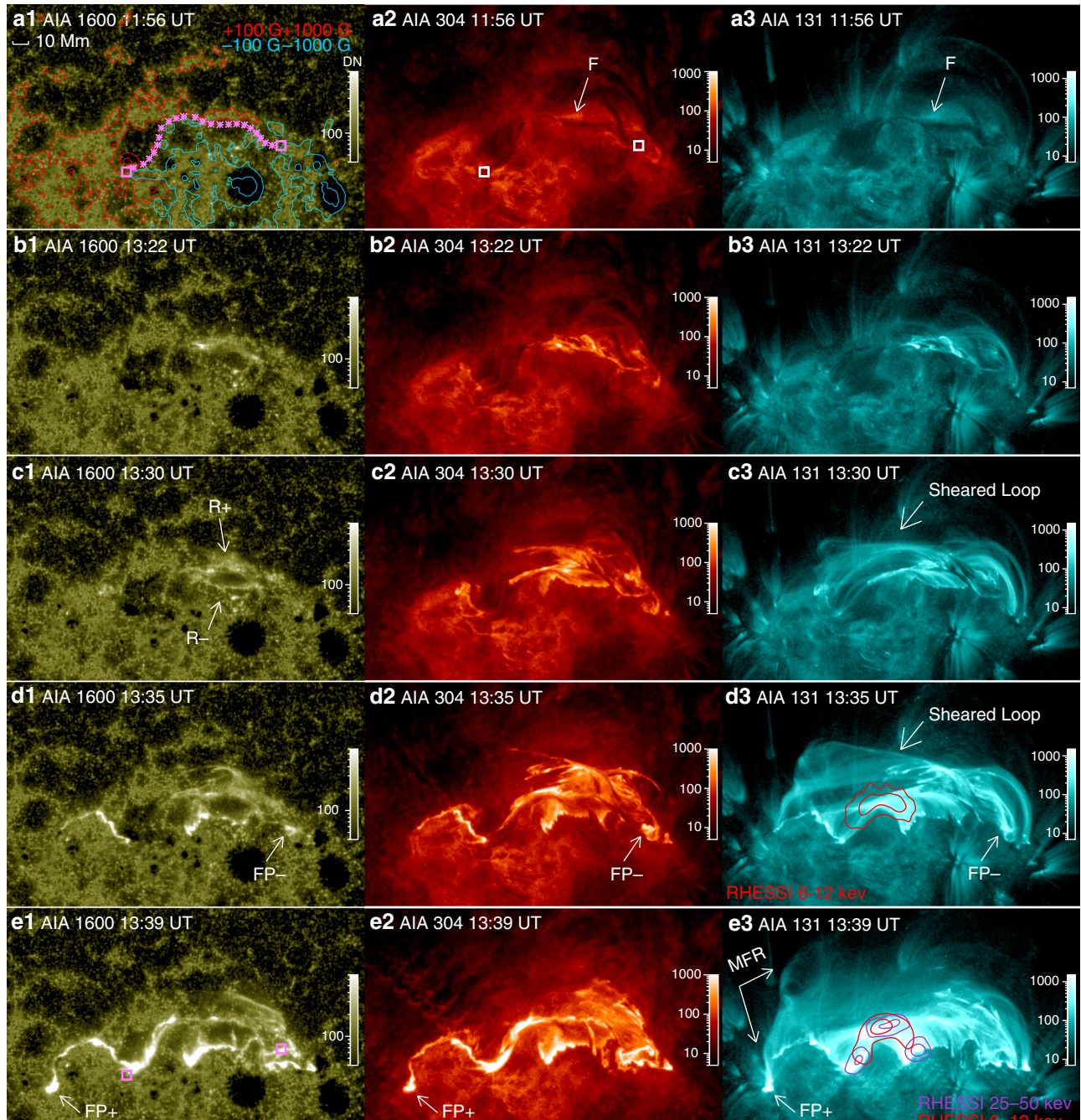

**Fig. 1** Snapshots of the solar eruption observed by the Atmospheric Imaging Assembly (AIA). From left to right column, the images in a logarithmic scale show the solar lower chromosphere in 1600 Å, upper chromosphere and transition region in 304 Å, and corona in 131 Å. Pixel values, i.e, the CCD data number (DN), are scaled by the color bar for each individual image. **a1** Is superimposed by the contours of the line-of-sight component of the photospheric magnetic field. A filament visible in 304 and 131 Å is located along the polarity inversion line, as marked by asterisks in **a1**, and its footpoints are marked by boxes. **d3**, **e3** Are superimposed by contours of 6–12 and 25–50 keV sources at the levels of 50 and 80% of the maximum brightness, observed by the Reuven Ramaty High-Energy Solar Spectroscopic Imager (RHESSI). The hard X-ray sources are reconstructed with the CLEAN algorithm[67]. The two feet of the magnetic flux rope (MFR) initially emerged as bright points (marked by arrows), with the western footpoint (FP−; **d1**–**d3**) appearing earlier than its eastern counterpart (FP+; **e1**–**e3**). An infant MFR formed at 13:39 UT in 131 Å, displaying in its eastern leg a fork-like feature that is anchored at FP+ **e3**

'R−') in both directions. The sheared loops were anchored on the ribbons, and the ribbon extension was accompanied by an apparent slipping motion of the loop footpoints along the ribbons (Fig. 1c3, d3; see also Supplementary Movie 3). The apparent slipping motion is a telltale sign of magnetic reconnection in 3D, when field lines exchange magnetic connectivities with their neighboring field lines in a QSL[28, 30, 35]. Besides the extension, the

two ribbons were also moving away from each other (Fig. 3b), a typical move in classic two-ribbon flares. By about 13:35 UT, the hot loops in 131 Å have developed into two groups: the high-lying sheared loops slipping toward the far ends of the flare ribbons, and the low-lying, less sheared flaring loops, which indicates that magnetic shear in the core field has been transported to higher altitudes via slipping magnetic reconnection. The HXR at this

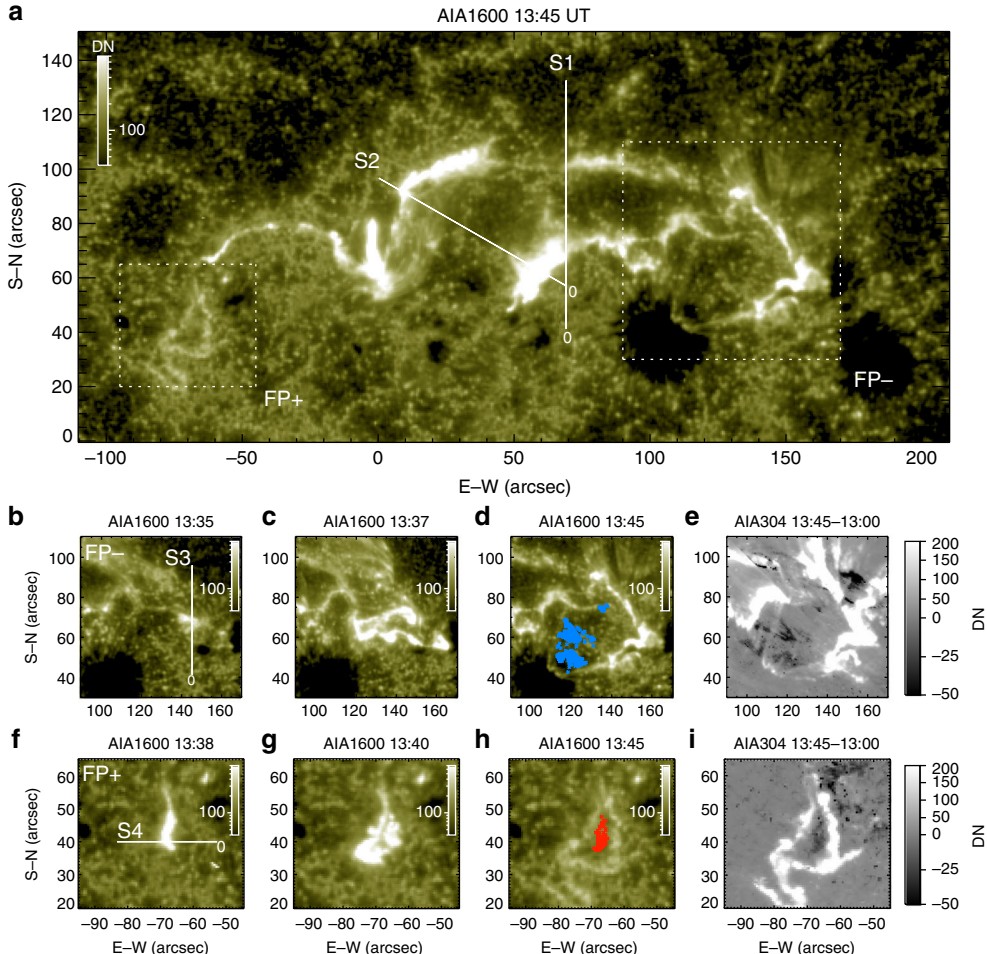

**Fig. 2** Formation and evolution of the magnetic flux rope's feet in 1600 Å. **a** Shows the morphology with two irregular bright rings (highlighted by rectangles) attached to the far ends of flare ribbons. The footpoint associated with negative polarity (FP−) is shown in **b**–**e** and that associated with positive polarity (FP+) in **f**–**i**. **e**, **i** Show dimming in 304 Å base-difference images. The dimmed pixels within FP− and FP+ are replotted in **d**, **h** in blue and red colors, respectively. Four virtual slits (S1–S4) are indicated in **a**, **b**, **f**, with their starting point marked by '0'

stage was mainly from thermal bremsstrahlung and co-spatial with the flaring loops (Fig. 1d3).

A sudden change of the ribbon morphology started at about 13:35 UT (marked by the second vertical dotted line in Fig. 3), when a new bright point appeared to the west of R− in AIA 1600 Å (Fig. 1d1) and soon expanded outward into an irregular bright ring attached to R− (Figs. 2b–e, 3d), defining a closed region (labeled 'FP−'). A few minutes later at about 13:38 UT, the eastern end of R+ bulged into a teardrop shape and also expanded outward into an irregular bright ring defining another closed region labeled 'FP+' (Figs. 2f–i, 3e). We note that the initial brightening of FP− is about 10 Mm away from the filament's western foot while that of FP+ is about 45 Mm from the filament's eastern foot. Coronal dimmings were observed to develop within FP+ and FP− almost simultaneously from about 13:41 UT onward (Fig. 2e, i), and persistent over the eruption period (Supplementary Fig. 2a, b). The dimmings were clearly visible and encompassed by similar bright rings in all of AIA's EUV passbands (Supplementary Fig. 2e–k and Supplementary Movie 2), suggesting that mass depletion is the dominant factor resulting in the observed dimming[36].

This new ribbon morphology, i.e., two bright rings attached to the far ends of flare ribbons (Fig. 2a), is observationally unprecedented in the literature, despite a few theoretical predictions[28, 29]. Its formation was associated with a topological transformation of the slipping sheared loops into an eruptive structure that drove a shocked wavefront ahead (Fig. 5 and Supplementary Movie 3). Particularly, two intertwining loops at the eastern leg of the eruptive structure constituted a fork-like feature (labeled 'MFR' in Fig. 1e3; see also Fig. 5), whose footpoint was nicely co-located with FP+ (Fig. 1e1–e3). Despite that the eruptive structure was undergoing rapid ascension and expansion, the fork-like feature was consistently present during 13:36–13:42 UT until it became too diffuse to be visible (Fig. 5), suggesting that it was a truly entangled structure rather than an illusion produced by projection effects or line-of-sight confusion; it was only seen in AIA 94 and 131 Å, indicating a temperature as hot as 6–10 MK. In the meantime, three HXR sources in the non-thermal energy range (25–50 keV) were observed, with two conjugate sources associated with R+ and R− and a third source located at the top of the thermal (6–12 keV) loop-like source (Fig. 1e3). The co-spatiality with the thermal loop top suggests that this non-thermal coronal source may result from coronal thick-target bremsstrahlung[37]. The segments of flare ribbons co-located with the HXR footpoints (crossed by a virtual slit S2 in Fig. 2a) moved away from each other at a faster speed (Fig. 3c) than the initial separation (Fig. 3b) as seen through a virtual slit S1 (Fig. 2a). By identifying brightened pixels in UV and dimmed pixels in EUV (Methods section; see also Supplementary Fig. 3 and Supplementary Movie 4), one can obtain magnetic fluxes

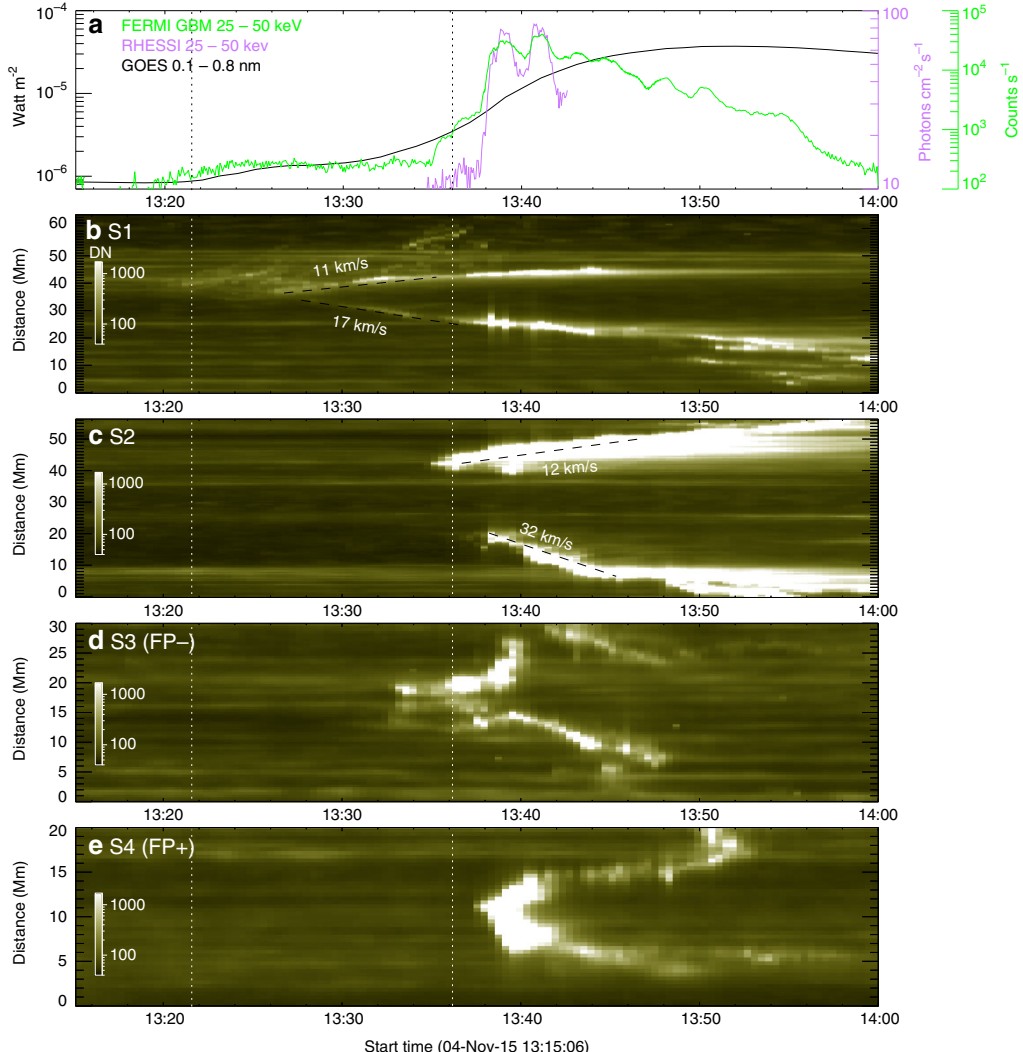

**Fig. 3** Temporal evolution of the flare ribbons in 1600 Å. **a** Shows 0.1–0.8 nm soft X-ray flux observed by the Geostationary Operational Environmental Satellite (GOES), 25–50 keV hard X-ray (HXR) count rate observed by the Gamma-ray Burst Monitor (GBM) onboard the Fermi Gamma-Ray Space Telescope, and 25–50 keV HXR photon flux observed by the Reuven Ramaty High-Energy Solar Spectroscopic Imager (RHESSI). **b–e** Show the evolution of the flare ribbons as seen through slits S1 and S2 indicated in Fig. 2a, and of the bright rings as seen through slits S3 (for FP−) and S4 (for FP+) indicated in Fig. 2b, f, respectively. The first vertical dotted line marks the onset of the flare with the initial appearance of flare ribbons; the second line marks the onset of the flare main phase characterized by rapid ribbon separation **c** and non-thermal HXR production **a**

(Supplementary Fig. 4) through the area swept by both flare ribbons and bright rings ($\Phi_R$), through the footpoint area enclosed by the bright rings ($\Phi_{FP}$), and through the dimming regions ($\Phi_D$).

Based on the above observations and multiple analysis techniques, we conclude that the observed MFR was dynamically formed during the course of the eruption, without a significant preexistence, because of the following evidence and reasoning.

First, from the perspective of morphology and evolution, the MFR's feet as identified by the conjugate coronal dimmings were fully enclosed by the irregular bright rings. Most importantly, each ring originated and expanded outward from a point-like brightening (Figs. 2b–i, 3d, e). This strongly suggests that the rope was being built up from almost none. As expected for a coherent MFR, the bright rings are interpreted as the footprint of a (quasi-)separatrix layer that wraps around the rope to separate twisted from untwisted field lines[1], a natural site for current concentration and dissipation[38].

Second, from the perspective of timing and location, the MFR's feet only began to emerge at the far ends of flare ribbons by the

end of the flare early phase as characterized by the ribbon extension and a gradual increase in SXRs, and grew in size during the flare main phase as characterized by the rapid ribbon separation and HXR bursts (Fig. 3). With the ribbons extending deeply into the flux concentrations of opposite polarities, the MFR's feet were located far away from the filament's, the latter of which had remained stationary in the neighborhood of the PIL prior to the eruption (Fig. 1 and Supplementary Note 1). Mapping the MFR's feet to a pre-eruption magnetogram, we found neither strong current density nor significant net current at either foot (Methods section; Supplementary Figs. 3b, 4c), in contrast to the results found at the footpoint regions of sigmoids[39].

Third, we detected no preexisting coherent MFR either in nonlinear force-free field modeling using two independent codes[40, 41] or a magnetohydrodynamic simulation of the coronal field employing the data-driven active region evolution (DARE) model[42] (Methods section; Supplementary Movie 5). This implies that the observed formation of the MFR is not driven by the local photospheric evolution, but likely triggered by nonequilibrium or instability of the coronal field.

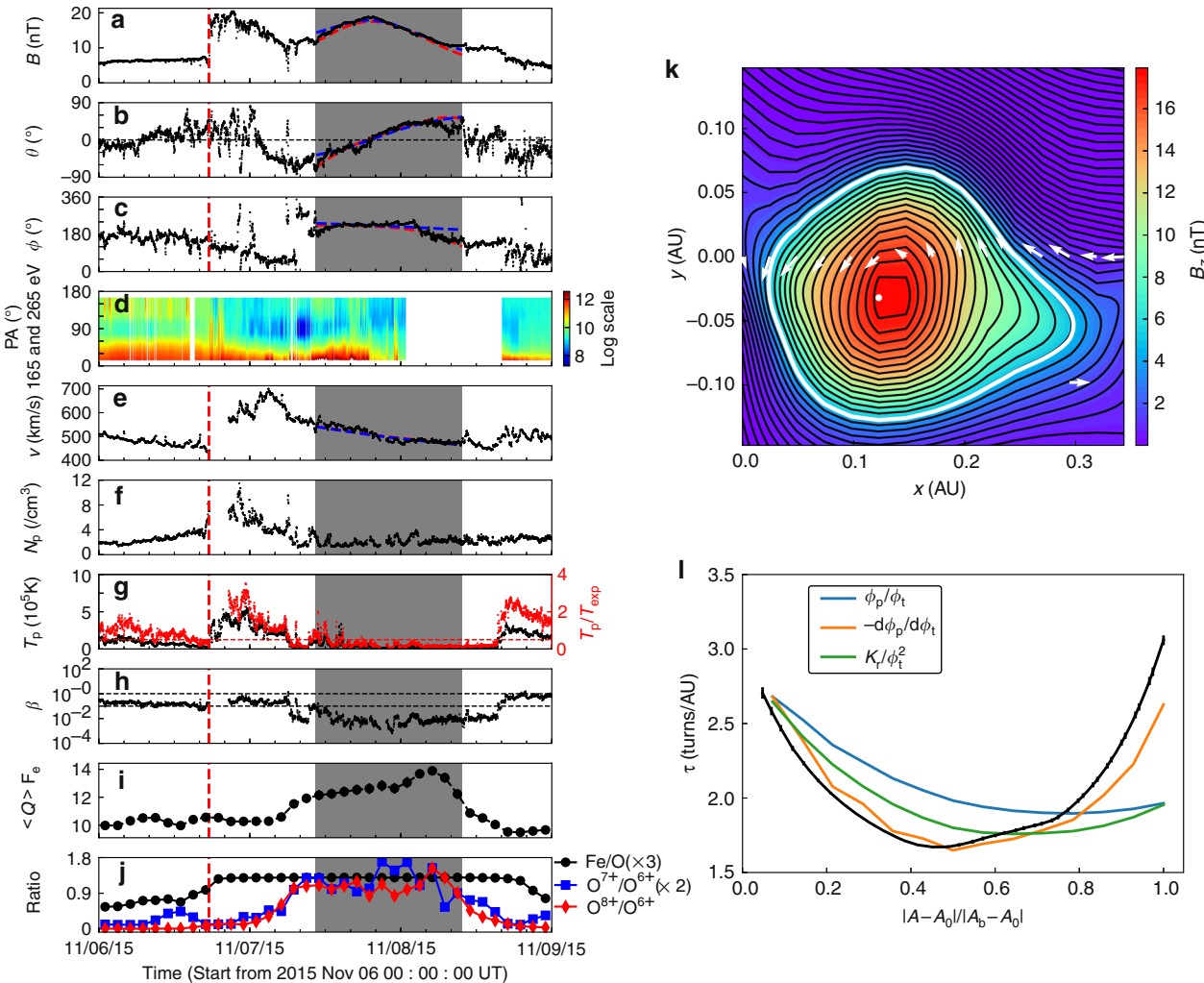

**Fig. 4** In situ diagnostics and analysis. **a** In situ observation of the magnetic cloud (MC; shaded region) preceded by a shock (red vertical line). All data except suprathermal electrons (WIND spacecraft) were obtained by the Advanced Composition Explorer (ACE) spacecraft. **a** Field magnitude $B$; **b** field inclination angle $\theta$ (with respect to the ecliptic plane); **c** azimuthal angle $\phi$ (0 deg pointing to the Sun), **d** pitch angle distribution of suprathermal electrons; **e** solar wind speed $V$; **f** proton density $N_p$; **g** temperature $T_p$ (superimposed by $T_p/T_{exp}$); **h** plasma $\beta$; **i** average charge states of iron $\langle Q \rangle_{Fe} = \sum_i Q_i n_i$ (density is normalized such that $\sum_i n_i = 1$), and **j** various composition ratios. The dashed lines in **a–c**, **e** show the model fitting results with red (blue) indicating the Gold-Hoyle (Lunquist) solution (Methods section). **k**, **l** Grad-Shafranov reconstruction (Methods section) of the MC. **k** Shows the reconstructed cross section traversed by ACE. The flux function $A(x, y)$ is represented by the black contours. The thick white contour marks the MC boundary $A_b = 38.9$ T·m. The strength of axial field $B_z(A)$ is indicated by colors, with the maximum marked by a white dot. The measured magnetic field vectors are projected along the spacecraft path $y = 0$ (white arrows). **l** shows the twist $\tau$ of MC field lines (black line) as a function of $|A - A_0|/|A_b - A_0|$. $A_0 = 160.4$ T·m gives the MFR center; also shown are $\Phi_p/\Phi_t$, $d\Phi_p/d\Phi_t$, and $K_r/\Phi_t^2$ (ref. [17])

Finally, the uniformly enhanced Fe/O inside the MC at 1 AU (Fig. 4j) argues for its coronal origin. Moreover, the average charge state of iron $\langle Q \rangle_{Fe}$ was increasingly elevated after entering the MC and decreasing upon exiting (Fig. 4i), which implies that the MFR experienced intense heating before leaving the corona[43, 44], consistent with the dynamic formation of the MFR in the corona. One caveat is that the spacecraft did not pass right through the MFR's center (Fig. 4k and Methods section), which leaves open a small chance that the core may posses a rather different charge state[44].

Furthermore, the newly discovered flare morphology and evolution opens an avenue for deciphering the buildup of magnetic twist, on which we will elaborate below. This process has not been carefully studied in numerical simulations, but can potentially provide additional testing/constraint on the models.

**Non-uniform twist profile**. The MFR is considered to be formally formed as a coherent structure only when both footpoints

came into being and were relatively fixed in position. Hence the magnetic flux through FP+, which appeared later but more stationary than FP- (Figs. 2, 3; Supplementary Movie 4), gives an accurate measurement of the MFR's toroidal (axial) flux $\Phi_t$ (Methods section and Supplementary Fig. 4). More importantly, the temporal variation of $\Phi_t$ not only indicates the growth of the MFR with time but also gives a glimpse of different shells within the rope. The MFR's poloidal flux $\Phi_p$ can be derived from the total reconnection flux, $\Phi_r \approx \Phi_p + \Phi_t$, which is the magnetic flux swept by flare ribbons. However, by counting brightened flare area in the chromosphere, what we obtained is $\Phi_R$, the flux swept by both the flare ribbons and the bright rings (Methods section). It is important to keep in mind that they represent footpoints of topologically distinct magnetic structures: the bright rings highlight the footpoints of newly reconnected field lines contributing to the burgeoning MFR, while the flare ribbons represent the footpoints of post-flare loops, but the former is only slightly dimmer than, and hence cannot be easily

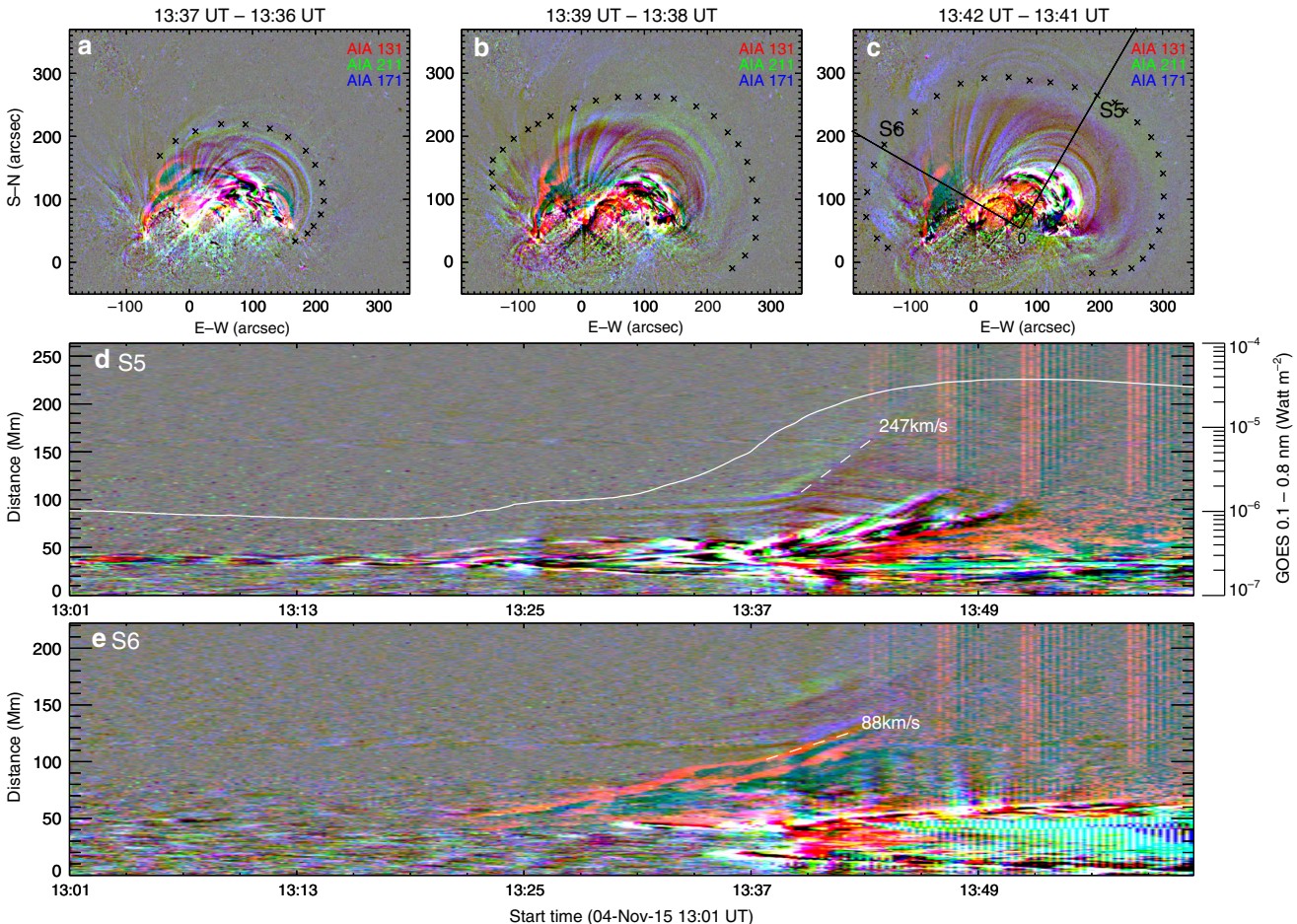

**Fig. 5** Eruptive structure in the solar eruption on 4 November 2015. **a–c** Show composite difference images in three Atmospheric Imaging Assembly (AIA) passbands, 171 (blue), 211 (green), and 131 Å (red). **d**, **e** Show the evolution of the eruptive structure as seen through the virtual slits S5 and S6 indicated in **c**. Linear fittings give the projected speeds of the features of interest (marked by dashed lines). The magnetic flux rope (MFR) is visible in 131 Å. Its eastern leg possesses a peculiar fork-like feature, which is shown as two semi-parallel red strips in the time-distance map **e** constructed via the silt S6 that crosses the fork. The wavefront (marked by crosses in **a–c**) ahead of the MFR is mainly visible in 171 and 211 Å, therefore appearing greenish

distinguished from, the latter. As a result, $\Phi_t$ is counted twice in $\Phi_R$. Hence, $\Phi_p \approx \Phi_R - 2\Phi_t$.

Now it is possible to derive the spatial distribution of magnetic twist within the MFR. Specifically, the ratio between the instant increments of $\Phi_p$ and $\Phi_t$, $\Delta\Phi_p(t)/\Delta\Phi_t(t)$, estimates the twist number at a certain shell, while $\Phi_p/\Phi_t$ measures the average twist across the rope. The resultant twist profile (Fig. 6c and Methods section) hence indicates that the field lines constituting the MFR's core were far more twisted than the average.

These results compare favorably with the characteristics of the interplanetary MC. We employed two independent techniques, force-free fitting and Grad-Shafranov (GS) reconstruction (Fig. 4), to estimate the magnetic flux in the MC (Methods section). We found that $\Phi_p$, $\Phi_t$, and $\Phi_p/\Phi_t$ of the solar MFR are comparable with those of the MC (Fig. 6c and Supplementary Table 1), respectively, and that the MC as reconstructed by the GS method also exhibits high twist in the center (~2.7 turns per AU, or, 5.4–8.5 turns given the cloud axial length ranging between 2 and $\pi$ AU), on which four different approaches agree (Fig. 4l). The MC's twist towards the boundary is less certain as the results reached by the different approaches diverge significantly.

## Discussion

Combining solar and interplanetary data, we concluded that the MFR possessed a highly twisted core, which cannot possibly be

formed in a quasi-stationary fashion prior to the eruption because such a high twist (up to 10 turns) falls far into the unstable zone of the helical kink instability. Recall that the early phase of the flare before the MFR's feet took shape was characterized by the flare-ribbon extension in association with the slippage of sheared coronal loops. This is reminiscent of the picture proposed by refs. [45, 46] as an effort to generalize the standard two-dimensional flare/CME model to the three dimensions. In this picture, the reconnection first occurs between a pair of sheared field lines to form a twisted field line with roughly one turn and an underlying less-sheared flare loop. This twisted field line continues to reconnect sequentially with adjacent sheared arcades, each reconnection injects some additional poloidal flux, yet roughly maintains its toroidal flux, which eventually forms a highly twisted infant MFR connecting the far ends of two flare ribbons.

Thus, we suggest that the instantaneous twist number reflects the frequency of reconnections between sheared field lines, with each reconnection adding roughly one turn into the twisted field line in formation. This is evidenced by the similarity in time profile (Fig. 6) between $\Delta\Phi_p(t)/\Delta\Phi_t(t)$, the non-thermal HXR emission (a proxy of both particle and CME acceleration[47]), and the time derivative of the reconnection flux $\Phi_r = \Phi_p + \Phi_t$ (a proxy of reconnection rate; Methods section). We further envisage that many such sequential reconnections take place transforming magnetic shear into twist, and the resultant twisted field lines congregate around the infant MFR in a self-organizing manner,

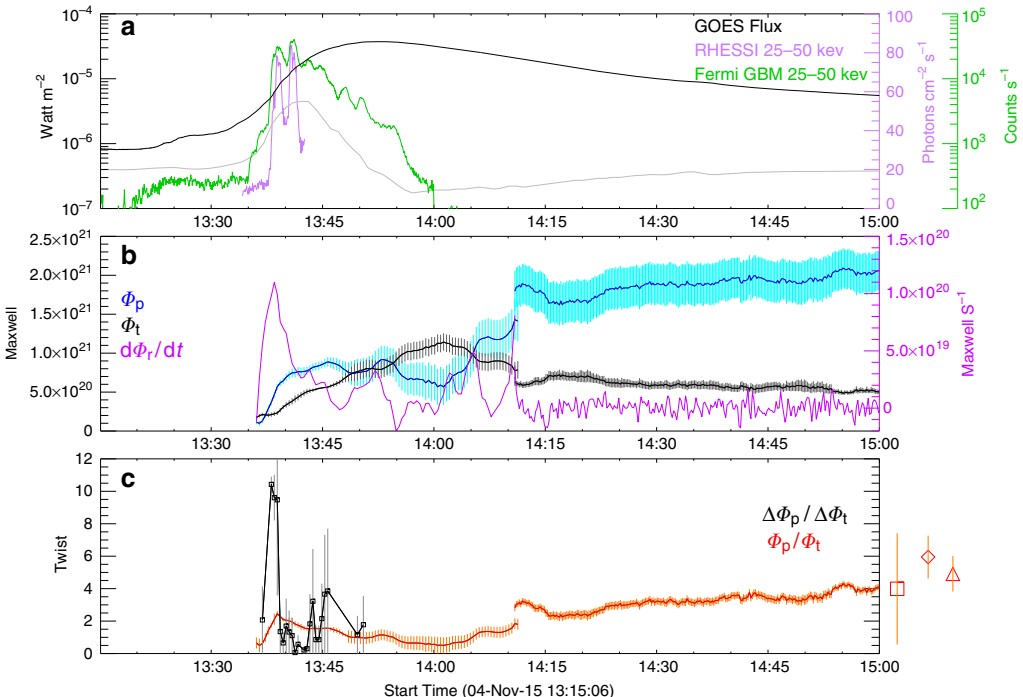

**Fig. 6** Temporal evolution of the poloidal and toroidal fluxes in the magnetic flux rope. **a** 0.1–0.8 nm soft X-ray flux observed by the Geostationary Operational Environmental Satellite (GOES), its time derivative in an arbitrary unit (gray), 25–50 keV hard X-ray (HXR) count rate observed by the Gamma-ray Burst Monitor (GBM) onboard the Fermi Gamma-Ray Space Telescope, and 25–50 keV HXR photon flux observed by the Reuven Ramaty High-Energy Solar Spectroscopic Imager (RHESSI). **b** Poloidal flux $\Phi_p$ and toroidal flux $\Phi_t$ of the magnetic flux rope. Also shown is the time derivative of reconnection flux $\Phi_r = \Phi_p + \Phi_t$. **c** Twist number of the magnetic flux rope as gauged by $\Phi_p/\Phi_t$ and $\Delta\Phi_p/\Delta\Phi_t$. Marked on the right are $\Phi_p/\Phi_t$ given by the Gold-Hoyle (square) and Lundquist (diamond) fittings and the Grad-Shafranov reconstruction (triangle; Methods section) of the interplanetary magnetic cloud, with the values and error bars given in Supplementary Table 1 (Methods section). Error bars of $\Phi_p$ and $\Phi_t$ are given by varying the detection threshold of the corresponding features (Methods section), and those of $\Phi_r$, $\Phi_p/\Phi_t$, and $\Delta\Phi_p/\Delta\Phi_t$ are derived following the error propagation rules

which enhances the MFR's Lorentz self force (also termed 'hoop' force[48]) as $\Phi_p$ increases, and hence facilitates its rising. As the MFR cuts through the overlying field, a positive feedback is established between its ascension and the magnetic reconnection, leading to the CME. The decrease in magnetic twist with time or from the core outward could also be a manifestation of magnetic reconnection progressing from strongly to less sheared flux, which is how the magnetic field around the flaring PIL is usually structured (see also Supplementary Movie 5).

Most recently, Priest and Longcope[49] linked two phases of reconnection, which is often observed in solar eruptions[50], to the buildup of magnetic twist: first the 3D zipper reconnection of sheared flux, which is associated with the extension of flare ribbons along the PIL, and then the quasi-2D main phase reconnection of unsheared flux around an MFR, which is associated with the separation of flare ribbons away from the PIL. Their conceptual model suggested that the zipper reconnection in a sheared arcade creates an MFR of ~1 turn, but substantial extra twist if starting with a preexisting MFR under the arcade. Either way, the subsequent main phase reconnection adds a layer of uniform twist of a few turns. In the current event, the evolution of flare ribbons and HXR emission also indicate two reconnection phases, except that the first phase displayed both extension and slow separation of flare ribbons. The MFR's feet only began to take shape at the beginning of the main phase as characterized by rapid ribbon separation and non-thermal HXR bursts (Fig. 3), with the highly twisted core produced at around the HXR peak (Fig. 6). Hence, the timing of high twist production seems at variance with this model. Obviously, it calls for three dimensional numerical modeling and simulation to understand the buildup of high twist by reconnection in the corona.

It is also worth noting that the filament may play an important role, serving as a trigger of the eruption and even as a seed upon which the eruptive MFR was built up. However, the eruptive MFR and the filament exhibit important differences in the following two aspects: first, the filament might be associated with an MFR twisted by ~1 turn, based on some filament activities prior to the eruption (Supplementary Note 1), but the eruptive MFR is highly twisted with a non-uniform twist profile; second, the filament's feet are relatively stationary in the neighborhood of the PIL (Supplementary Figs. 8, 9), but the eruptive MFR's feet is dynamically formed during the flare main phase, in places far away from the PIL. This poses a great challenge to the solar eruption models that rely on a preexisting MFR, whose footpoints are assumed to be anchored on the dense photosphere all the time.

## Methods

**SDO data processing.** SDO/AIA is equipped with seven EUV narrow-band channels spanning a broad range of temperature sensitivities, i.e., 131 Å (Fe XXI for flare, peak response temperature log $T = 7.05$; Fe VIII for AR, log $T = 5.6$)[51] 94 Å (Fe XVIII, log $T = 6.85$), 335 Å (Fe XVI, log $T = 6.45$), 211 Å (Fe XIV, log $T = 6.3$), 193 Å (Fe XXIV for flare, log $T = 7.25$; Fe XII for AR, log $T = 6.2$), 171 Å (Fe IX, log $T = 5.85$), and 304 Å (He II, log $T = 4.7$); and two UV passbands, i.e., 1600 Å (C IV, log $T = 5.0$) and 1700 Å (continuum).

The evolution of AR 12,443 was monitored by the Helioseismic and Magnetic Imager (HMI) onboard SDO. The vector magnetograms are disambiguated and deprojected to the heliographic coordinates with a Lambert (cylindrical equal area; CEA) projection method, resulting in a pixel scale of 0.03° (or 0.36 Mm)[52]. The flow field on the photosphere is obtained by applying the differential affine velocity estimator for vector magnetograms (DAVE4VM)[53] to the time-series of vector magnetograms.

**Configuration and evolution of the active region.** When the M3.7 flare occurred on 4 November 2015, AR 12,443, located at the disk center and well isolated from

other active regions, was under slow decay with diminishing magnetic flux and electric current (Supplementary Fig. 1), In spite of displaying a semi-$\delta$ configuration with the negative polarities intruding into positive polarities, it was generally quiet and only produced two M-class flares during its disk passage, M1.0 on 31 October 2015 and M3.7 on 4 November, respectively, and became quite dormant afterward. This is consistent with the lack of photospheric features that are usually associated with eruptive behaviors, e.g., flux emergence or systematic shearing/twisting flows, except for some converging flow toward where the filament is located (Supplementary Fig. 1b). This converging flow is likely responsible for the activation of the filament (Fig. 1). The cancellation of magnetic flux brought together by converging flows toward the PIL is considered in some models to explain the filament formation[11] or the CMEs occurring during the decaying phase of an active region[54, 55].

To understand the magnetic connectivities within the active region, we built a nonlinear force-free field (NLFFF) to approximate the coronal field, using the code package developed by T. Wiegelmann[40, 56]. To best suit the coronal force-free condition ($\mathbf{J} \parallel \mathbf{B}$), the pre-flare vector magnetogram is preprocessed[57] before being taken as the extrapolation boundary. We further calculated the distribution of squashing factor $Q$[58] and twist number $\mathcal{T}_w$[1] in the NLFFF, but failed to identify a coherent MFR, which would otherwise be spotted as a volume of enhanced twist number ($|\mathcal{T}_w| \geq 1$) bounded by high-$Q$ surfaces. This result is confirmed by an independently developed NLFFF code, which has been demonstrated to be capable of recover an MFR in the weak field region[41].

To gain further insight, we employed the DARE model[42] to simulate the dynamic evolution of the coronal field. DARE has successfully reproduced eruptions with[42] or without a CME[59]. It solves a full set of time-dependent MHD equations with the bottom boundary continuously driven by photospheric vector magnetograms. A projected characteristic method is used to ensure a self-consistent coupling between the evolving coronal field and the driven surface field. An extrapolated NLFFF[41] is taken as the initial condition, and the plasma is simply adiabatic. Here the simulation is started 24 h before the onset of the flare (13:00 UT on 3 November 2015) by feeding a time sequence of vector magnetograms at 12 min cadence into the model, and is terminated until 12 h after. The modeling volume is a box of $512 \times 512 \times 512$. The result is shown in Supplementary Movie 5. One can see that the coronal magnetic configuration displays no significant changes over the 36 h period. Particularly, no significantly twisted field lines or coherently concentrated currents are found in the simulation.

**Identification of flare ribbon and coronal dimming**. Flare ribbons, including the bright rings in our case, are detected in the AIA 1600 Å passband by counting all the brightened pixels within the active region. Through a trial-and-error approach, an optimal threshold value is set to be certain times of the median value in a quiet region, so that we can pick up as many pixels in the flare ribbons as possible while diminishing as much as possible the interference of the eruptive structure and bright plages around the sunspots (Supplementary Movie 4). The threshold value is varied by ±10% to provide an uncertainty of the detection.

The two dimming regions are visible in all seven EUV channels of AIA (Supplementary Fig. 2 and Supplementary Movie 2). The 335 and 304 Å passbands give the best visibility and are least interfered by coronal loops. We hence carried out the detection in both passbands. We count all the pixels with brightness below a threshold within the identified boundary of the dimming regions. The threshold, again found by a trial-and-error approach, is a fraction of the pre-flare value within the boundary, and similarly the uncertainty is estimated by varying the threshold by ±10%. During the impulsive phase, the boundary of dimming is identified with the bright ring by edge detection algorithms. During the gradual phase, however, the dimming regions are only partially bounded by emission (Supplementary Movie 3). We hence used the earlier complete boundary as the reference, assuming that the dimmed segment of the boundary remained stationary, while performing the real-time detection of the brightened segment which was still evolving.

We then projected the identified pixels onto a pre-flare $B_z$ map with the same CEA projection method (Supplementary Fig. 3), and calculated signed magnetic fluxes through the brightened ($\Phi_{R+}$ and $\Phi_{R-}$; Supplementary Fig. 4e) and dimmed ($\Phi_{D+}$ and $\Phi_{D-}$; Supplementary Fig. 4d) areas. To get a sense of the change rate of $\Phi_R$, we obtained $d\Phi_{R+}$ and $d\Phi_{R-}$ by integrating the signed magnetic flux through the newly brightened area in each 1600 Å image. The signed fluxes through the MFR footpoints, $\Phi_{FP+}$ and $\Phi_{FP-}$ (Supplementary Fig. 4b) are approximated as those through the areas within the bright rings before about 14:10 UT and the dimmed areas afterward, using 304 Å images throughout for consistency. It is worth noting that the peak values of $\Phi_{FP+}$ and $\Phi_{FP-}$ are almost equal to each other.

We calculated the signed current through FP+ and FP− (Supplementary Fig. 4c) based on the same pre-flare $B_z$ map at 13:00 UT. We found that the electric current is roughly balanced and that there is no strong current density at either foot, in contrast to significant net currents and occasionally strong current density found at the footpoint regions of sigmoids[39]. This is consistent with our interpretation that the MFR was mainly formed during the eruption without a significant preexistence. Alternatively the net current can be calculated by integrating the transverse field component along the bright rings. The results from both approaches are in rough agreement.

In this study, magnetic fluxes calculated with $B_z$ are almost the same as those with the line-of-sight magnetogram because the flare occurred nearly at the disk

center with minimal projection effects, but generally the CEA projection should yield more accurate results.

**Estimation of poloidal and toroidal flux of the MFR**. The heating of the lower atmosphere during flares, which is observed as bright flare ribbons, is an immediate response to the energy deposit along field lines from the site of magnetic reconnection in the corona, since the timescales of magnetic reconnection, energy transport, and heating of the lower atmosphere (a fraction of a second to a few seconds) are much shorter than the cooling time of flare ribbons in the upper chromosphere or transition region (several minutes). The evolution time scale of the photospheric magnetic fields is even longer, typically hours to days. Due to flux conservation, a connection between the coronal magnetic field undergoing reconnection and the lower-atmosphere field at the energy deposit site is established as follows[16, 60],

$$\frac{\partial \Phi_r}{\partial t} = \frac{\partial}{\partial t} \int B_c \, dS_c = \frac{\partial}{\partial t} \int B_n \, dS_n, \tag{1}$$

where $\Phi_r$ is the magnetic reconnection flux and $\partial \Phi_r / \partial t$ gives the magnetic reconnection rate. $\Phi_r$ is defined by the integration of the inflow magnetic field $B_c$ at the reconnection site over the reconnection area $S_c$ in the corona. How to measure coronal magnetic field has been a long-standing problem, hence we obtain $\Phi_r$ by integrating the normal component of the magnetic field $B_n$ over the brightened flare area $S_n$ in the lower atmosphere, the latter of which reflects the instant footprints of separatrices along which magnetic reconnection takes place.

Conventionally the MFR's magnetic flux is decomposed into poloidal flux $\Phi_p$ and toroidal flux $\Phi_t$. In cylindrical symmetry,

$$\Phi_p = \int_0^R \int_0^{L_z} B_\phi(r) \, dr \, dz, \tag{2a}$$

$$\Phi_t = \int_0^R \int_0^{2\pi} B_z(r) \, r \, dr \, d\phi. \tag{2b}$$

Based on the solar observations, we conclude that the MFR mainly formed during the course of the flare, thus the total reconnection flux must account for all the magnetic flux in the MFR, i.e., $\Phi_r \approx \Phi_p + \Phi_t$, where $\Phi_t$ is identified with the areas encompassed by the bright rings attached to the far ends of the flare ribbons (FP+ and FP−; Fig. 2). Within the rings coronal dimmings (D+ and D−) were developed subsequently. FP+ and FP− give a more accurate depiction of the footpoint areas than D+ and D−, especially during the impulsive phase, as can be seen from Fig. 2 and Supplementary Fig. 4. However, the bright ring of FP+ formed several minutes later than FP−. It is a reasonable assumption that the MFR formally took shape only when both footpoints came into being and became relatively fixed in position. Thus, $\Phi_{FP+}$ follows more closely the development of $\Phi_t$. On the other hand, when one counts the brightened pixels in 1600 Å to obtain $\Phi_R$, one cannot differentiate between the flare ribbons and the bright rings, the latter of which also swept through the footpoint areas (Supplementary Fig. 3a). In effect, $\Phi_r \approx \Phi_R - \Phi_t$, hence we may further derive that

$$\Phi_p \approx \Phi_R - 2\Phi_t. \tag{3}$$

We have taken two slightly different approaches to calculate the ratio between the instant increments of $\Phi_p$ and $\Phi_t$, i.e., $r = \Delta\Phi_p(t)/\Delta\Phi_t(t)$. For the first approach, the increment is given by neighboring data points, i.e., $r = (\Phi_p(i+1) - \Phi_p(i))/(\Phi_t(i+1) - \Phi_t(i))$. The time difference between $i + 1$ and $i$ is determined by the AIA data cadence (12 or 24 s), but negative values are discarded. The result is shown in Fig. 6c. Alternatively, when either term is negative, we move on to the next data point, until both $\Phi_p(j) - \Phi_p(i)$ and $\Phi_t(j) - \Phi_t(i)$ are positive, where $j = i + 1, i + 2, ....$. The time at the midpoint between $j$ and $i$ is taken as the time of the resultant ratio. The two approaches yield similar time profiles (Supplementary Fig. 5), but only the first approach gives similar twist numbers as the in situ results. Note as the flare progressed into the decay phase, $\Delta\Phi_p$ and $\Delta\Phi_t$ became smaller and more noisy. Accordingly their ratio became more uncertain. Only during the impulsive phase, both $\Phi_p$ and $\Phi_t$ increased rapidly and the results are relatively robust.

**Estimation of magnetic flux in the MC**. We used two state-of-the-art techniques to estimate the magnetic flux in the interplanetary MC. In the first, we employed the velocity-modified cylindrical flux rope model[61], which improves upon the traditional force-free fitting method[19] by taking into account the in situ measurements of velocity vectors to model the dynamical evolution of MCs. The MC of interest is fitted with both a linear force-free Lundquist solution with increasing twist from the axis to the boundary[61] and a nonlinear force-free Gold-Hoyle solution with uniform twist[32], assuming a locally cylindrical symmetry. The cloud can be fitted fairly well by both solutions (Fig. 4a–c, e), yielding very similar results (Supplementary Table 1). From both fittings, the closest distance between the ACE spacecraft path and the rope axis is about $0.3R_c$, where $R_c \sim 0.2$ AU is the radius of the MFR's cross section. In particular the poloidal flux is estimated by assuming that the MC's axial length lies in the range $2-\pi$ AU[61].

In the second, we carried out the standard GS reconstruction[62, 63], assuming a translation symmetry along the flux-rope axis, i.e., $\partial/\partial z \simeq 0$. In the transverse $xy$ plane, magnetized plasma in quasi-static equilibrium is governed by the GS

equation,

$$\frac{\partial^2 A}{\partial x^2} + \frac{\partial^2 A}{\partial y^2} = -\mu_0 \frac{dP_t}{dA} = -\mu_0 j_z(A), \qquad (4)$$

where $P_t = p + B_z^2/2\mu_0$ is the sum of plasma pressure and axial magnetic pressure. The transverse field $(B_x, B_y)$ is completely determined by the flux function $A(x, y)$ as follows, $B_x = \partial A/\partial y$ and $B_y = -\partial A/\partial x$. Field lines completing at least one full turn are projected as closed contours of $A$ in the $xy$ plane.

A flux rope solution for the present MC is obtained with a fitting residue $R_f = 0.05$ and a boundary $A = A_b$[64] (Supplementary Fig. 6). The thick white contour in Fig. 4k represents a flux surface enclosing the core structure corresponding to $A > A_b$, where the model assumptions, two-dimensionality and quasi-static equilibrium, are deemed well satisfied based on the fitting. The solution exhibits a typical MFR structure with nested flux surfaces of monotonically decreasing axial field from the center (Fig. 4k). Within the flux surfaces field lines wind around the axis $z$ in a right-handed sense. Various physical quantities can be derived from the GS reconstruction results[16, 17, 65]. In particular, the toroidal (axial) and poloidal flux, all evaluated for the volume within the boundary $A = A_b$, are $0.919 \times 10^{21}$ Mx and $1.754 \times 10^{21}$ Mx (per AU), respectively. Assuming the same range for the MC's axial length as in the force-free fittings above, we estimate the total poloidal flux $(4.509 \pm 1.002) \times 10^{21}$ Mx. A similar range, 2–4 AU[65], has been obtained by comparing the GS reconstruction with the lengths of magnetic field lines derived from in situ energetic electrons. Both poloidal and toroidal fluxes are roughly the same as those derived by the force-free fittings, and comparable to those derived from solar observations in orders of magnitude (Supplementary Table 1).

The twist of a field line $\tau(A)$ is given by

$$\tau(A) = \frac{1}{L_z}, \qquad (5)$$

where $L_z$ in units of AU is the distance from the root of the field line on the $z = 0$ plane to the point where a full turn is completed. Field lines rooted at the same $A$ contour should yield the same $L_z$. A distribution of $\tau$ as a function of $A$ (black curve in Fig. 4l) is hence obtained by calculating $L_z$ for the sampled root points on all the closed contours of $A$. A caveat to keep in mind is that this method is more reliable near the center than near the boundary[17]. Also shown are results from three other methods to approximate $\tau$ for cylindrical MFRs using magnetic fluxes ($\Phi_p$ and $\Phi_t$) and relative helicity $K_r$, namely, $\Phi_p/\Phi_t$, $d\Phi_p/d\Phi_t$, and $K_r/\Phi_t^2$ (see the Appendix in ref. [17]). Combining the four methods, we conclude that the MC possesses a highly twisted core, i.e., ~2.7 turns per AU, or, 5.4–8.5 turns, given the uncertain axial length of the MC. Its twist toward the boundary is less certain as the results given by the four methods diverge.

Projecting the MC axis orientations derived from the models onto the solar surface, we found that the MC rotated clockwise within 30° with respect to the MFR orientation on the Sun, which is approximated by the connection of its footpoints (Supplementary Fig. 7). The clockwise rotation is consistent with the MC's positive helicity sign[66] as derived by the MC models, suggesting that the conversion of twist into writhe was ongoing for the MFR due to the helical kink instability. The small rotation angle is expected when an MFR rises and expands faster than the development of the kink instability.

**Data availability**. Raw data are available from the corresponding spacecraft missions. Derived data supporting the findings of this study are available from the corresponding author upon reasonable request.

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

## Acknowledgements

R.L. acknowledges the support by NSFC 41774150, NSFC 41474151 and the Thousand Young Talents Program of China. Y.W. acknowledges the support from NSFC 41774178 and 41574165. C.S acknowledges the support from NSFC 41774181. This work was also supported by NSFC 41421063, CAS Key Research Program of Frontier Sciences QYZDB-SSW-DQC015, and the fundamental research funds for the central universities.

## Author contributions

R.L. led the investigation, interpreted the data, and wrote the manuscript. W.W. processed and analyzed the AIA data under R.L.'s guidance. Y.W. performed the force-free fitting of the MC. Q.H. performed the GS reconstruction of the MC. C.S. processed and analyzed in situ data. C.J. carried out the data-driven simulation of the active region. C.Z. made preliminary analysis of the AIA data. All authors participated in discussion and made contribution to finalize the manuscript.

## Additional information

**Competing interests:** The authors declare no competing financial interests.

