## [Peer Review File · Nature Communications]

Reviewers' comments:

Reviewer #1 (Remarks to the Author):

This paper describes and analyzes an interesting solar eruption observed during initiation at the Sun and as an interplanetary CME at 1 AU. By carefully measuring the time-dependent emission and photospheric magnetic-field characteristics, and assuming well-established relationships between the flare ribbons and reconnected flux, the authors draw several important conclusions. First, this eruption did not begin with a preexisting flux rope, but rather with a sheared arcade. Second, the flux rope formed during the eruption appears to be highly twisted at its core and less twisted with radial distance from the axis. Third, the interplanetary CME contains at least twice the magnetic flux of the initial flux rope, indicating that significant reconnection must have occurred long after the flare ribbons became invisible.

The paper is well-written and gives adequate credit to previous studies of solar eruptions. The analysis appears thorough and well-motivated, with most assumptions clearly stated out. I disagree with some of the interpretations (see below), but overall this excellent research offers significant information that could enhance our understanding of the CME/eruptive flare initiation process. This work is well worth publishing after some revisions.

Besides the comments listed below, the main shortcoming of this paper is the failure to use this information to discriminate among the competing models for solar eruptions. Models relying on preexisting flux ropes (e.g., the torus instability model) are clearly ruled out by this event, while reconnection-based models (e.g., the breakout model) are consistent with the observations and analysis. It would add appreciably to the value of this paper if it contained a discussion of the implications for the models. I would be happy to see a revised version of this manuscript, as well as the authors' responses to the issues raised below.

The following questions, comments, and suggestions are given in terms of the page, paragraph, and line. Major issues are in boldface, minor ones are in regular font.

1) pg. 1, para. 1 after Abstract: the first sentence is rather redundant; all helices contain helical field lines.

2) pg. 3, para. 2, line 11: the loops are not necessarily slipping, so it would be more accurate to say "an apparent slipping motion..." here. The whole concept of "slipping reconnection" is controversial and could be a distraction from the valid points in this paper.

3) pg. 4, para. 2, line 11-12: please explain how a nonthermal HXR source could be associated with a cool, dense prominence. Prominences are observed frequently to heat up during eruption, but the hotter regions are not localized at the loop top source, to the best of my knowledge.

4) pg. 4, para. 3, line 6: the year should be 2015 not 2016.

5) pg. 5, first line: "possess" is missing the final "s"

6) pg. 5, para. 2, line 8: should say "the absence of a preexisting coherent MFR..."

7) pg. 6, para. 2: the twist distribution in the MFR also could be strong evidence for reconnection progressing from sheared to unsheared flux, as predicted by some models and consistent with the nature of filament-channel fields. It would be useful to refer to more recent simulation studies that predict flux rope characteristics formed during eruption, not just the cartoon pictures of the Gosling references.

8) pg. 7, para. 3: it is unclear why hoop force is singled out as the cause of the MFR rising. The positive feedback mentioned here is a key feature of the breakout model for CME initiation, among others.

9) pg. 11, para. 3 (under Eq. 1): it is unclear what “higher than the in-situ speed..” refers to. The ambient solar wind speed, the magnetic cloud speed, or something else? A few words of explanation are needed.

10) pg. 13, last para., line 9: what is meant by “the spacecraft path” --- ACE and WIND paths?

11) pg. 14, para. 3: considering the large shear in the interior of the flux rope, is this configuration stable, e.g., to kinking?

Figures: In general, the small thin print, fine lines, and compressed labels are difficult to see. (e.g., Fig. 6a and Suppl. Fig. 4)

Fig. 1: the contours of the LOS photospheric field in Panel a1 are invisible to me.

Fig. 2: the dimmed pixels in (d) and (h) are very difficult to see, particularly the blue ones.

Fig. 3: I think the last sentence in the caption is backwards. Fig. 2 shows MFR footpoints not flare ribbons. Please check this.

Fig. 4: The GOES emission measure is not visible in panel (a) --- I do not see any red lines, but there is a grey line there as described in the caption but not in the panel key.

Fig. 5: The wavefront is hard to see in panels (a)-(c), and does not appear to me to be yellow.

Suppl. Fig. 1: what is the maximum vector length in the middle panel?

Suppl. Fig. 2: Can't read the bottom key (D-, in green) against the grey background. Perhaps a white box behind the key would help?

Suppl. Fig. 5: Why is this Figure included? The reference in the text (pg. 4) talks about the spatial locations of the emission (illustrated in Fig. 1), not the spectra.

Reviewer #2 (Remarks to the Author):

The paper presents interesting results on manifestations of a magnetic flux rope (MFR) in the solar corona and interplanetary space. The authors used multiwavelength coronal images and photospheric vector magnetic field measurements, as well as in-situ data on solar wind parameters. MHD modeling was performed in order to reconstruct the interplanetary magnetic cloud structure. In spite of a number of interesting and important findings, several inferences seem unfounded or too hyperbolic.

The main conclusion that the observed MFR was formed during the course of the solar eruption (which is reflected in the title of the paper) is not substantiated enough. While the authors insist on the formation of the MFR from a sheared arcade, their results and presented data show the reverse.

1. The initial structure of coronal threads observed in AIA images looks more like a twisted rope than a sheared arcade. This appearance could be illusive but the motion of bright feature from the north side of the filament to the south side between 13:00 UT and 13:08 UT (Movie 2) shows the different shift of a field line and most probably corresponds to the bottom part of the helical field line. This is the evidence of the presence of the MFR before the eruption.

2. Two-(or more)-fold difference between the flux estimations in the magnetic cloud (MC) and in the reconnection areas also testifies to the presence of the significant flux additional to the reconnection flux.

Thus, the title of the paper and the authors' interpretation of results seem very doubtful.

The estimation of the radial speed of the dome-like wavefront (envelope) is also too strained. Obviously the front moves not radially, nearly along the line-of-sight, as the authors claim. The structure expands in all three dimensions, so the speed in the sky-plane is close to the radial speed. Moreover, the general direction of motion of the envelope is most likely not radial but deviates from sunspots as it is often observed. This is confirmed also by the field line pattern presented in Movie 4. Therefore, the radial speed of the dome-like envelope should be much less than 1500 km/s, which will be more consistent with the average speed of the MC propagation from the Sun to 1 AU (800 km/s).

Reviewer #3 (Remarks to the Author):

I commend the authors on carrying out a detailed and well-integrated study of an interesting solar eruptive event and corresponding magnetic cloud. That being said, I am concerned that this work does not provide more than an incremental advance in the study of flux rope formation in the corona. As I detail below, examples of the key observations reported in this manuscript (e.g., footpoint identification and dimming, dynamic flux rope formation) have already been published during the past 25 years of research on magnetic clouds and their solar sources. Thus, while the observations reported here are compelling, its not clear that these findings constitute more than an incremental confirmation of existing knowledge. Given this reading of the manuscript, I am requesting that the authors respond with a clear explanation of (1) what this manuscript contributes in terms of NEW understanding of flux rope formation and propagation in the corona; and (2) why these contributions constitute more than an incremental confirmation of existing knowledge.

To be more specific regarding the above comments, my biggest concern is the claim that the identification of the two flux rope footpoints and their dimming is unprecedented. The authors conspicuously fail to cite the work of Webb et al. [2000], which presents essentially the same type of observations and conclusions, albeit from more primitive observational platforms (EIT/MDI instead of AIA/HMI). In spite of the lower-quality observations available in 2000, the conclusions of the Webb et al. paper are strikingly similar to those of the present manuscript. As such, it is

paramount that the authors delineate how the present manuscript constitutes a substantial advance beyond previous work.

With regard to dynamic flux rope formation, Cheng et al. [2011] used AIA to report on a dynamic formation event. As such, this is also not the first manuscript to report AIA observations of flux rope formation. I only point this out to help the authors identify the unique contributions of their manuscript in their reply.

In conclusion, I am happy to review the detailed points of this manuscript if the authors are able to make a compelling case that the manuscript provides more than an incremental confirmation of existing knowledge and is therefore suitable for publication in Nature Communications.

Replies to the Referees:

We thank the three reviewers who made very helpful comments that benefit us greatly in revising the paper. We have made efforts to improve the manuscript by addressing all the points raised. Below are the point-to-point replies to the comments, typed in Arial Black font. Major changes of text in the manuscript are shown in blue.

Reviewer #1 (Remarks to the Author):

This paper describes and analyzes an interesting solar eruption observed during initiation at the Sun and as an interplanetary CME at 1 AU. By carefully measuring the time-dependent emission and photospheric magnetic-field characteristics, and assuming well-established relationships between the flare ribbons and reconnected flux, the authors draw several important conclusions. First, this eruption did not begin with a preexisting flux rope, but rather with a sheared arcade. Second, the flux rope formed during the eruption appears to be highly twisted at its core and less twisted with radial distance from the axis. Third, the interplanetary CME contains at least twice the magnetic flux of the initial flux rope, indicating that significant reconnection must have occurred long after the flare ribbons became invisible.

The paper is well-written and gives adequate credit to previous studies of solar eruptions. The analysis appears thorough and well-motivated, with most assumptions clearly stated out. I disagree with the some of the interpretations (see below), but overall this excellent research offers significant information that could enhance our understanding of the CME/eruptive flare initiation process. This work is well worth publishing after some revisions.

Besides the comments listed below, the main shortcoming of this paper is the failure to use this information to discriminate among the competing models for solar eruptions. Models relying on preexisting flux ropes (e.g., the torus instability model) are clearly ruled out by this event, while reconnection-based models (e.g., the breakout model) are consistent with the observations and analysis. It would add appreciably to the value of this paper if it contained a discussion of the implications for the models. I would be happy to see a revised version of this manuscript, as well as the authors' responses to the issues raised below.

Reply: We are grateful to this constructive comment and have included a few remarks in the concluding section on the implication of the presented observations for the models. It is our opinion, however, that the debate on the pre-eruption configuration mostly originates from ambiguity in observation, and that numeric models can adapt quickly once the ambiguity is resolved. For example, the breakout model would

have no problem in accommodating a preexisting rope, while the torus instability is still relevant for a newly formed rope, in that whether it will successfully erupt or not is regulated by the background magnetic field. Hence, we did not put heavy emphasis on discriminating among the models on the basis of their initial configurations, but focused on the buildup of magnetic twist, which is disclosed by our new analysis technique, and can potentially provide additional testing / constraint on the models.

The following questions, comments, and suggestions are given in terms of the page, paragraph, and line. Major issues are in boldface, minor ones are in regular font.

1) pg. 1, para. 1 after Abstract: the first sentence is rather redundant; all helices contain helical field lines.

Reply: we have revised the sentence as “A magnetic flux rope consists of helical magnetic field lines.....”

2) pg. 3, para. 2, line 11: the loops are not necessarily slipping, so it would be more accurate to say “an apparent slipping motion...” here. The whole concept of slipping reconnection” is controversial and could be a distraction from the valid points in this paper.

Reply: revised as suggested.

3) pg. 4, para. 2, line 11-12: please explain how a nonthermal HXR source could be associated with a cool, dense prominence. Prominences are observed frequently to heat up during eruption, but the hotter regions are not localized at the loop top source, to the best of my knowledge.

Reply: We found this northern HXR coronal source is similar (cospatiality with the thermal loop top) to the events interpreted as coronal thick-target bremsstrahlung, which is often attributed to the presence of dense coronal material. We have revised the sentence accordingly.

4) pg. 4, para. 3, line 6: the year should be 2015 not 2016.

5) pg. 5, first line: “possess” is missing the final “s”

6) pg. 5, para. 2, line 8: should say “the absence of a preexisting coherent MFR..”

Reply: 4-6 have been corrected as suggested.

7) pg. 6, para. 2: the twist distribution in the MFR also could be strong evidence for reconnection progressing from sheared to unsheared flux, as predicted by some models and consistent with the nature of filament-channel fields. It would be useful to refer to more recent simulation studies that predict flux rope characteristics formed during eruption, not just the cartoon pictures of the Gosling references.

Reply: We are grateful to this very thoughtful comment, and have included it in the text . However, to our best knowledge, few simulations have been dedicated to studying the buildup of magnetic twist either prior to or during eruptions, or, produced a spatial twist distribution of the flux rope involved in the simulation. Priest and Longcope (2017, Solar Physics) made the most up-to-date relevant progress. Their highly simplified model illuminates the basic processes involved, but still at variance with the current observation in various aspects (last paragraph of the main text).

8) pg. 7, para. 3: it is unclear why hoop force is singled out as the cause of the MFR rising. The positive feedback mentioned here is a key feature of the breakout model for CME initiation, among others.

Reply: We agree that hoop force is not likely the only cause of the MFR rising, and have hence softened the tone in the text. It may play a significant role because of the high twist in the core. This is left for a future detailed investigation in collaboration with modelers. We refrain from referring to the breakout model because it is not clear that either a quadrupolar field or a coronal null is involved in this event.

9) pg. 11, para. 3 (under Eq. 1): it is unclear what “higher than the in-situ speed..” refers to. The ambient solar wind speed, the magnetic cloud speed, or something else? A few words of explanation are needed.

Reply: The cited value (550 km/s) is for the interplanetary shock.

10) pg. 13, last para., line 9: what is meant by “the spacecraft path” --- ACE and WIND

paths?

Reply: ACE spacecraft, because of a gap in WIND data, which can be seen from the pitch-angle distribution of suprathermal electrons.

11) pg. 14, para. 3: considering the large shear in the interior of the flux rope, is this configuration stable, e.g., to kinking?

Reply: The flux rope is under fast expansion and propagation, hence the stability issue may be irrelevant. It is left for future investigation whether internal interaction / reconnection could occur in a flux rope with uneven twist distribution.

Figures: In general, the small thin print, fine lines, and compressed labels are difficult to see. (e.g., Fig. 6a and Suppl. Fig. 4)

Fig. 1: the contours of the LOS photospheric field in Panel a1 are invisible to me.

Fig. 2: the dimmed pixels in (d) and (h) are very difficult to see, particularly the blue ones.

Reply: We have modified the figures for better visualization.

Fig. 3: I think the last sentence in the caption is backwards. Fig. 2 shows MFR footpoints not flare ribbons. Please check this.

Reply: Thanks for spotting this error. It has been corrected.

Fig. 4: The GOES emission measure is not visible in panel (a) --- I do not see any red lines, but there is a grey line there as described in the caption but not in the panel key.

Reply: Annotations in Fig.4 have been corrected.

Fig. 5: The wavefront is hard to see in panels (a)-(c), and does not appear to me to be yellow.

Reply: The wavefront is now marked by crosses to aid the visualization; the color is indeed not yellow, but greenish, because of strong response in 211 (green) and 171 (blue), yet weak response in 131 (red).

Suppl. Fig. 1: what is the maximum vector length in the middle panel?

Reply: The maximum vector length is denoted by the arrow at the lower left corner.

Suppl. Fig. 2: Can't read the bottom key (D-, in green) against the grey background. Perhaps a white box behind the key would help?

Reply: The intersection of D+ and R is shown in cyan, and that of D- and R in yellow.

Suppl. Fig. 5: Why is this Figure included? The reference in the text (pg. 4) talks about the spatial locations of the emission (illustrated in Fig. 1), not the spectra.

Reply: This figure is meant to show the thermal and nonthermal energy ranges of the HXR spectra, but has been removed due to its minor relevance to the current study.

Reviewer #2 (Remarks to the Author):

The paper presents interesting results on manifestations of a magnetic flux rope (MFR) in the solar corona and interplanetary space. The authors used multiwavelength coronal images and photospheric vector magnetic field measurements, as well as in-situ data on solar wind parameters. MHD modeling was performed in order to reconstruct the interplanetary magnetic cloud structure. In spite of a number of interesting and important findings, several inferences seem unfounded or too hyperbolic.

The main conclusion that the observed MFR was formed during the course of the solar eruption (which is reflected in the title of the paper) is not substantiated enough. While the authors insist on the formation of the MFR from a sheared arcade, their results and presented data show the reverse.

1. The initial structure of coronal threads observed in AIA images looks more like a twisted rope than a sheared arcade. This appearance could be illusive but the motion of bright feature from the north side of the filament to the south side between 13:00 UT and 13:08 UT (Movie 2) shows the different shift of a field line and most probably corresponds to the bottom part of the helical field line. This is the evidence of the presence of the MFR before the eruption.

2. Two-(or more)-fold difference between the flux estimations in the magnetic cloud (MC) and in the reconnection areas also testifies to the presence of the significant flux additional to the reconnection flux.

Thus, the title of the paper and the authors' interpretation of results seem very doubtful.

Reply: The only plausible explanation for the observational fact that both footpoints of the eruptive MFR evolved from a point-like brightening after the flare onset is that the MFR is mainly formed during the eruption. A seed flux rope if existing prior to the flare must be too small to be resolved by AIA, otherwise the initial footpoint brightening would be ring-like rather than point-like.

We did take into account the possibility that the filament could be associated with a preexistent rope; but even so, its field must have undergone significant changes before being incorporated into the eruptive MFR, because the former's footpoints were distinctively separated from the latter's (Figure 1; 1st paragraph in the concluding section). The formation process could include the incorporation of the filament field through magnetic reconnection with the surrounding sheared arcade, to account for the footpoint displacements between the filament and the eruptive MFR.

We do not agree that the apparent motions and shapes in EUV can be considered as a substantial evidence for the presence of a preexisting flux rope because of projection effects of these 3D structures and line-of-sight confusion in the optically thin corona, which could be illusive, as rightfully noted by the reviewer.

A natural explanation for larger fluxes in the MC than those derived from flare ribbons is that magnetic reconnections that serve to build up the MFR persist longer than flare ribbons, since the growth of post-flare arcades in EUV typically continues after the disappearance of flare ribbons. However, the estimate of fluxes in magnetic clouds is model-dependent and subject to large uncertainties. Hence one can only safely conclude that they are comparable to those derived from solar observations in orders of magnitude, which is how previous studies approached this issue (Qiu et al. 2007; Hu et al. 2014). We have revised the text for clarification.

The estimation of the radial speed of the dome-like wavefront (envelope) is also too strained. Obviously the front moves not radially, nearly along the line-of-sight, as the authors claim. The structure expands in all three dimensions, so the speed in the sky-plane is close to the radial speed. Moreover, the general direction of motion of the envelope is most likely not radial but deviates from sunspots as it is often observed. This is confirmed also by the field line pattern presented in Movie 4. Therefore, the radial speed of the dome-like envelope should be much less than 1500 km/s, which will be more consistent with the average speed of the MC propagation from the Sun to 1 AU (800 km/s).

Reply: We agree that the wavefront ahead of the erupting MFR is supposed to be dome-like, moving in all directions, but instead it looks like a loop in EUV in the low corona (Figure 5 and Supplementary Video 2), which corresponds most likely to the top section of the dome in projection, due to the effect of line-of-sight integration from an oblique viewpoint from above. We surmise that the top section of the dome moves with a significant velocity component towards the observer, because the location is very close to the disk center and its projected speed in the low corona is only ~250 km/s, one order of magnitude smaller than the measured Alfvén speed in active regions (Aschwanden 2005). We hence estimated that the speed of the coronal wave lies in the range [1050, 1500] km/s. The lower limit matches the average speed of the shock propagating from the Sun to the Earth (800 km/s), if the shock decelerated linearly to 550 km/s at 1 AU, which is a highly unlikely situation because coronal waves typically decelerate rapidly in the low corona (Warmuth 2015, Living Rev. Solar Phys.). The higher limit is given by the simplified projection correction, which assumes that the wavefront moves radially at a uniform speed in the low corona. The real speed would be smaller if there is a deviation from the radial direction because of projection effects. This speed range is comparable with the characteristic Alfvén speed in active regions and those of EUV fast coronal waves observed during the SDO era. Nevertheless, we have made it clear in the text that it is only a crude estimation, without taking into account the detailed distribution of local Alfvén speed in the active region.

The observation of a Type II radio burst recorded by the Bruny Island Radio Spectrometer lends support to our estimation. The burst drifted from 190 to 115 MHz during 13:42 - 13:44 UT, which gives an average shock speed of 1100 km/s with the two-fold Newkirk density model.

Reviewer #3 (Remarks to the Author):

I commend the authors on carrying out a detailed and well-integrated study of an interesting solar eruptive event and corresponding magnetic cloud. That being said, I am concerned that this work does not provide more than an incremental advance in the study of flux rope formation in the corona. As I detail below, examples of the key observations reported in this manuscript (e.g., footpoint identification and dimming, dynamic flux rope formation) have already been published during the past 25 years of research on magnetic clouds and their solar sources. Thus, while the observations reported here are compelling, it's not clear that these findings constitute more than an incremental confirmation of existing knowledge. Given this reading of the manuscript, I am requesting that the authors respond with a clear explanation of (1) what this manuscript contributes in terms of NEW understanding of flux rope formation and propagation in the corona; and (2) why these contributions constitute more than an incremental confirmation of existing knowledge.

Reply: We appreciate that the reviewer raised a critical question as to whether the current study provides more than an incremental confirmation of existing knowledge. Our answer is affirmative, as nothing similar has been reported in the literature to our best knowledge. The justification in detail is given below.

The presented observation makes an unambiguous case for a newly formed flux rope during the course of the eruption. This formation process is disclosed with unprecedentedly clarity from a morphology and evolution of flare ribbons that have not been noticed before. This observation even allows us to resolve quantitatively the buildup of magnetic twist in a flux rope, and to reveal its intimate relation to flare reconnection. In particular, the buildup of high twist (up to 10 turns) in the core, which is consistent with the radial twist profile of the interplanetary MC, has never been reported before from either solar observations or numeric models and simulations of solar eruptions, but this process can potentially provide additional testing/constraint on the models. For clarification, we list below the major findings in the current study, in comparison with previous studies.

Current study	Previous studies
MFR feet ENCLOSED by irregular bright rings attached to far ends of two flare ribbons	MFR anchored within the OPEN hooks of two J-shaped flare ribbons (e.g., Javier et al. 2014)
Bright rings defining the MFR feet originate from point-like brightenings, indicating MFR is newly formed during the eruption	N/A

Well-defined dimming developed within, and fully enclosed by, bright rings hosting the MFR feet	Diffuse dimming without a clear boundary or a clear relation to flare ribbons (e.g., Webb et al. 2000; Qiu et al. 2007; Hu et al. 2014)
Highly twisted core and less twisted outer shells of the MFR are produced during the flare impulsive phase in the low corona	N/A for solar observation; but some MCs possess similar radial twist profiles (Hu et al. 2014)
Buildup of twist correlated with HXR lightcurve and reconnection rate	N/A

To be more specific regarding the above comments, my biggest concern is the claim that the identification of the two flux rope footpoints and their dimming is unprecedented. The authors conspicuously fail to cite the work of Webb et al. [2000], which presents essentially the same type of observations and conclusions, albeit from more primitive observational platforms (EIT/MDI instead of AIA/HMI). In spite of the lower-quality observations available in 2000, the conclusions of the Webb et al. paper are strikingly similar to those of the present manuscript. As such, it is paramount that the authors delineate how the present manuscript constitutes a substantial advance beyond previous work.

Reply: We disagree that Webb et al. (2000) presented "essentially the same type of observations and conclusions". Webb et al. did report that the flare arcade in their case was flanked by adjacent dimmings, but the dimming regions were very diffuse and exhibited no clear boundary, only marked subjectively by hand (their Figure 3). The relation between dimmings and flare ribbons was not clear, nor was investigated by Webb et al. The eastern ribbon did show a hook-like structure on its northern end (their Figure 2, left bottom panel), but this hook looks much smaller than the corresponding dimming region (their Figure 3, right top panel). It is unlikely that the flux rope in Webb et al. (2000) was newly formed, and they did concluded that it originated "in a large coronal structure linked to the erupting filament". It is also unlikely that similar observations had been missed before due to "more primitive observational platforms", because 1) the well-defined dimming regions and their emitting boundaries in our case are observed by all of SDO/AIA EUV passbands, including 171 and 193 Å, which have very similar coverage and responses as 171 and 195 Å passbands of SOHO/EIT (spatial resolution ~ 4"), STEREO/EUVI (spatial resolution ~ 2") and TRACE (spatial resolution ~ 1", similar to AIA); and 2) the emission enclosing the dimming regions in our case is also observed by

AIA 1600 Å, which is similar to TRACE 1600 Å. Therefore, we provided an unambiguous observation and analysis of well-defined dimming developed within, and fully enclosed by, bright rings that define the feet of an MFR under formation, which has not been achieved before.

With regard to dynamic flux rope formation, Cheng et al. [2011] used AIA to report on a dynamic formation event. As such, this is also not the first manuscript to report AIA observations of flux rope formation. I only point this out to help the authors identify the unique contributions of their manuscript in their reply.

Reply: Cheng et al. (2011) presented an observation of a bright blob of hot plasma above the limb, which was believed to be a flux rope "still in the formation phase in the low corona". It is not clear from their observation whether the flux rope is preexistent or newly formed, as stated by the authors, "we believe that the blob of the hot plasma, which later formed the full-fledged flux rope through magnetic reconnection, originated from either a sheared core field in the low corona or a weakly twisted flux rope existing prior to the eruption". Most importantly, the interpretation of such coronal features suffers inevitably from projection effects and line-of-sight confusion in the optically thin corona, and hence seldom unambiguous. Also their analysis involved no magnetic field measurements nor in-situ confirmation. In contrast, we concentrate on the morphology and evolution of flare ribbons, which is not subject to projection effects nor line-of-sight confusion; and our analysis has a strong support from in-situ diagnostics.

In conclusion, I am happy to review the detailed points of this manuscript if the authors are able to make a compelling case that the manuscript provides more than an incremental confirmation of existing knowledge and is therefore suitable for publication in Nature Communications.

Webb, D. F., R. P. Lepping, L. F. Burlaga, C. E. DeForest, D. E. Larson, S. F. Martin, S. P. Plunkett, and D. M. Rust. "The origin and development of the May 1997 magnetic cloud," J. Geophys. Res. 105 27251 (2000).

Cheng, X., J. Zhang, Y. Liu, and M. D. Ding. "Observing flux rope formation during the impulsive phase of a solar eruption." Astrophys. J. Lett. 732 L25 (2011).

Reply: We appreciate that the reviewer pointed out the above two relevant references. We have included both in the revised manuscript.

To summarize, the presented observation is unprecedented and of unambiguous clarity, and our analysis technique is innovative. This

study has achieved a significant advance in understanding the formation and eruption of flux ropes, a topic of fundamental importance in solar and heliospherical physics, by resolving the formation process and twisting distribution of the flux rope from a morphology and evolution of flare ribbons that has not been noticed before. This will certainly motivate future observation, modeling, simulation, as well as laboratory experiment.

Reviewers' comments:

Reviewer #1 (Remarks to the Author):

2nd Review of "Formation of a Highly Twisted Magnetic Flux Rope During the Course of A Solar Eruption" by Wang et al.

The authors have responded satisfactorily to the comments and criticisms of the reviewers, including myself, added clarifications and corrections to the text, and made the figures easier to understand. In my opinion the paper can be accepted for publication. I have a few suggestions for corrections to fix grammatical or typographical errors, and a couple of comments intended for the authors only (no changes needed in the text).

- 1) The first sentence of the abstract should begin with "The"
- 2) "heliospheric" is misspelled in line 4 of the abstract (h and p are reversed)
- 3) "feet" should be "foot" in line 6 of the abstract
- 4) pg. 4, para. 3, line 3: as per my previous review, "slipping" should have "apparently" before it
- 5) pg. 5, para. 3, lines 5-7: the sentence beginning with "A 'seed'" is confusing. Here is a suggested rewrite (ignore italics): If a 'seed' MFR existed prior to the flare, it must have been too small to be resolved by AIA; otherwise the initial footpoint brightening would be ring-like rather than point-like.
- 6) pg. 6, para. 1, and pg. 7, last para.: "numeric" should be "numerical"
- 7) pg. 6, para. 3, line 4: replace "to the 3D occasion" by "to three dimensions"
- 8) pg. 12, last line: the projection could be either toward or away from the observer. Either would reduce the projected speed.
- 9) pg. 13, para. 3: replace "a highly unlikely situation" by "under the unlikely assumption"
- 10) pg. 26, Table 1: "Shafranov" is misspelled.
- 11) In the response to my review, the authors remark that, in their opinion, the debate on pre-eruption configuration mostly originates from ambiguity in observations. I would point out that there are significant differences among models in terms of relative timings of key phases (e.g., flare reconnection vs CME fast acceleration) and underlying magnetic field (e.g., breakout requires a multipolar configuration, whereas the ideal instability and tether cutting models do not). On the other hand, all models agree that a flux rope exists after the eruption starts, so post-eruption signatures are not that useful for testing/constraining the models.

Reviewer #2 (Remarks to the Author):

The paper was not modified enough to change my opinion. I still think that the authors' interpretation of results is not the only possible and is not substantiated enough. The title may mislead a reader that the paper presents the strong evidence of formation of the MFR exclusively during the eruption. However, it is only one of possibilities following from the observations.

I cannot agree that the initial flux-rope footpoint brightening should be ring-like rather than point-like. Every brightening starts from separate points and then spreads into threads, ribbons, etc.

Dense filament material does not necessarily fill the whole length of a flux rope or other magnetic structure supporting it. It is usual that the ends of an eruptive filament are rather far from previously visible ends of the undisturbed filament. The filament in NOAA 12443 evolved significantly before the eruption. About 07 UT, four hours before the eruption, there were two close filaments within the same filament channel with different endpoints, as it is clearly visible in H-alpha images of Kanzelhoehe Observatory. At late phases of an eruption, many filaments show magnetic relation to remote areas of the solar surface. It can be the manifestation of coronal

reconnection or activation of coronal loops (e.g., Grechnev et al., 2008, Solar Phys. 253, 263; van Driel-Gesztelyi et al., 2014, ApJ 788, 85). Anyway, these processes do not reveal the formation of flux ropes.

The authors easily threw off my pointing to the evidence of the presence of the MFR before the eruption, while they insist on the presence of a not more evident sheared arcade.

The authors still consider the motion of a loop-like structure in EUV images as radial and make wrong correction of the projected speed. They admit that the structure of the wavefront ahead of the erupting MFR is supposed to be dome-like and the loop appears in EUV images due to the effect of line-of-sight integration. In this case with the nearly radial direction of the line-of-sight, the optical density is maximum on a nearly vertical side of the dome but not at the top section. Thus the observed loop moves rather more transverse the line-of-sight than along it as supposed the authors.

Reviewer #3 (Remarks to the Author):

First, given the authors' response to my previous request to differentiate this work from the previous results of Webb et al., I am willing to grant that the detailed analysis of the magnetic fluxes in the corona and at 1 AU constitutes a substantial advance in the analysis of a dynamically formed flux rope and its associated magnetic cloud. That being said, for the reasons I outline below, I find that a major revision is still required in order for this paper to be suitable for publication.

=====
=====

After repeated readings of this manuscript, the following structure is my best understanding of the authors' intent for this work:

1. The flux rope forms dynamically in the corona:
 - a. Footpoints are defined by bright rings that start as point-like brightenings
 - b. Subsequent dimmings within the bright footpoint rings indicate mass drainage
 - c. No detection of a coherent flux rope before the eruption (NLFFF, DARE)
 - d. Minimal net footpoint current when mapped to the pre-eruption J_z
2. The flux rope has a non-uniform twist profile with a highly twisted core:
 - a. Time-dependent flux accounting in the corona (brightened pixel tracking)
 - b. In situ observations of the resulting magnetic cloud (ACE, WIND)
 - c. Total flux comparison between corona and 1 AU (multiple methods)
 - d. Twist profile comparison between corona and 1 AU (GS reconstruction)

(I understand there may be a third conclusion, which is that the magnetic cloud has more flux than the observed coronal rope, but the authors need to determine if there is enough evidence to warrant treating this as a stand-alone conclusion or whether to fold it into the analysis of the twist profile.)

In the above structure, there are two key results, each of which is supported by four 'observations and analysis' bullet points. Frankly, the present layout of the paper does a poor job of conveying these results to the reader. In fact, many of the salient bullet points listed above are buried in the methods section and barely referenced in the main body. Conversely, the main body contains some ancillary information that doesn't help the reader understand the key results. While this ancillary information (such as the details of the magnetic cloud charge states) is important, it can be moved to the methods section in order to clear the way for more information in the main body

that directly pertains the key results. Given these considerations, I am requesting that the authors undertake a major structural revision of this manuscript in order to make the key results more accessible to the reader.

=====
=====

Abstract:

The authors assume that the reader implicitly understands that "in situ spacecraft measurements" means measurements acquired at 1 AU (L1). This is not obvious to the general reader and should be stated in such a way that the two are explicitly connected.

Starting with "Here we identify..." every sentence should be aimed at the two key results of the paper. Something like, "Here we identify the dynamic formation of a flux rope during a solar eruptive event..."

The sentence beginning with "These results suggest..." tells the reader that the results in the previous sentences support the formation of a MFR with a highly twisted core. But the previous sentence only talks about the bright footpoint rings and the dimmings. The results that support the formation of a MFR with a highly twisted core are the flux accounting measurements, which are not directly mentioned here. They need to be mentioned in the abstract in order for the conclusions to make sense.

Finally, the abstract does not give enough credence to the analysis of the magnetic cloud at 1 AU. A full sentence in the abstract should be dedicated to the 1 AU analysis (GS solution, twist profile).

=====
=====

Introduction:

In paragraph 2, I refute the authors' assertion that solar eruptions are believed to be governed by a single physical process. As the authors themselves state in their reply to the reviewers, a given event could be influenced by breakout reconnection, the torus instability, or a combination of the two. As such, multiple physical processes can clearly be involved.

The last paragraph before the results should be expanded to give a more definitive outline of the paper (see below). Also, 'in situ diagnostics' should be something more like 'in situ measurements of the resulting magnetic cloud at 1 AU.'

=====
=====

Results section:

I recommend a restructuring of the subheadings in this section as described below.

Present structure:

- > Observations and Analysis
- > Conclusions and Discussion

Suggested new structure:

- > Observations of the 4 Nov 2015 eruptive event (corona + 1 AU)

- > Dynamic flux rope formation (corona)
- > Non-uniform twist profile (corona + 1 AU)
- > Discussion

The 'Observations' subheading should be a condensed version of the first few 'Results' paragraphs, and could also include a paragraph on the in situ observations at 1 AU. Ancillary information like the details of the different AIA channels and the charge state observations of the magnetic cloud can be mentioned, but the bulk of the content should be moved to the Methods in order to condense this section and clear the way for more results (see below).

The next two subheadings should focus on conveying the 1a-1d and 2a-2d bullet points from above. For example, under the 'Dynamic formation' subheading, bullet point 1d on minimal net footpoint current should be described with a direct reference to the relevant data-driven modeling in the Methods section. At present, this result, which is one of the most important arguments against the existence of a pre-eruption flux rope, is not even mentioned in the main body of the paper.

As presently constructed, the main body does not do justice to the conclusion that the flux rope is believed to have a non-uniform twist profile with a highly twisted core. The entirety of this description is in the paragraph that begins with "The MFR is considered to be formally formed..." This description should be expanded and should include more information on the comparison between the $\Delta \Phi_p / \Delta \Phi_t$ coronal measurements and the twist profile in the GS reconstruction of the magnetic cloud. This a central result in the paper that should be appropriately emphasized.

Finally, substantial effort (and one of the six main figures) is devoted to analysis of the EUV shock wave and propagation speed of the eruption. Maybe I'm missing something, but this doesn't seem to be crucial to either of the key conclusions of the paper. Is it necessary to include this analysis? It seems like ancillary information at best. Consider moving it all to the Methods section.

A small note, but the plasma beta should be properly defined in the text.

=====

Main figures:

Figure 3:

The virtual slits (S1-S4) could be better labeled so that the reader doesn't have to flip back and forth. For instance S3 -> S3 (FP-) and S4 -> S4 (FP+).

Figure 4:

Why does the x-axis cover such a long time range when all of the action is packed in the 13:30-14:30 time window? You could cut it off at 15:00 rather than 16:00 so that the reader can better examine the critical time window. Also, how much should we trust the early-time $\Delta \Phi_p / \Delta \Phi_t$ points that give the high central twist result? I know this is crucial to your conclusions, but it would seem to depend highly on how the footpoints (and therefore the toroidal flux) are defined early in time. As you state toward the end of the paper, caution is warranted when interpreting this result. I'll come back to this point in the discussion of the GS reconstruction of the MC in Figure 6.

Figure 5:

As discussed above, the shock wave analysis could be moved to Methods since it doesn't add directly to the dynamic formation or non-uniform twist results. Does this contribute substantially enough to the dynamic flux rope formation result to warrant keeping as a main figure?

Figure 6:

This figure is barely addressed in the main text, but it is a key component of the non-uniform twist profile conclusion. With regard to the twist profile, the readout in Fig. 6c of the twist from the GS solution of the MC structure is key. Unfortunately, some important information is missing from the x-axis. In particular, there is no marking of the flux rope boundary, $|A_b - A_0|$. If I examine Supplementary Fig. 6 in detail, it looks like $A_0 \sim 155 \text{ T}\cdot\text{m}$ and $A_b \sim 40 \text{ T}\cdot\text{m}$. Does this mean that the x-axis cutoff in Fig. 6c at $|A - A_0| \sim 120 \text{ T}\cdot\text{m}$ is the flux rope boundary at $|A_b - A_0|$? As is common in the fusion community, one could consider plotting the normalized flux $|A - A_0| / |A_b - A_0|$, which would give a normalized x-axis running from 0 to 1.

If $|A_b - A_0| \sim 120 \text{ T}\cdot\text{m}$, then the edge of the rope is as twisted as the core with a lower twist region in between. Why, then, do the $\Delta \Phi_p / \Delta \Phi_t$ observations in Fig. 4c not show a rise in $\Delta \Phi_p / \Delta \Phi_t$? It appears this latter injection phase is not tracked by the coronal measurements. The authors should flesh out these considerations for the reader.

I think the result that such high twist is observed in the flux rope without a kink instability has interesting implications for the timescales. As the authors point out in their reply to the reviewers, the fast expansion of the highly twisted rope must nullify the impact of the kink instability. This could be stated more precisely in terms of the relevant timescales. The kink instability cannot arise faster than the Alfvén time along the rope. If the propagation itself is quasi-Alfvénic, then the kink instability won't have time to develop.

=====
=====

Methods section:

The NLFFF and DARE modeling should be explicitly referenced in the main body of the text in support of bullet points 1c and 1d.

With regard to the projected speed of propagation, I find this subsection to still be confusing, and, as outlined above, I question its relevance to the paper. Consider revising further. For one, it's not clear to me where the new 1050 km/s lower bound comes from. There are half a dozen speeds listed in just a few sentences, and I can't tell what some of them mean and/or how they are related.

Be sure to emphasize that the B_z mapping of ribbon and footpoint fluxes is derived from the pre-flare B_z map. This was not clear to me at first. I was confused about the zero net current comment until I realized it was zero net current before the flare.

I can't seem to rationalize the flux accounting in Eq. 4. In particular, the factor of 2 in the $2\Phi_t$ term seems to result from circular logic. How is the total reconnected flux different from the total ribbon flux? It seems to me that the math is simpler: $\Phi_r = \Phi_R = \Phi_t + \Phi_p$ such that $\Phi_p = \Phi_R - \Phi_t$ (without the factor of two). Please try to clarify how the factor of two comes into the equation. Is there another reference that uses this flux accounting formulation? The factor of two clearly ripples through all of the interpretation of the poloidal flux, twist, flux ratios, etc. (e.g., Supplementary Table 1), so it is important to resolve this apparent contradiction.

List the values of A_0 and A_b from the GS reconstruction, in accord with Fig. 6 discussion.

=====
=====

Supplementary figures:

Supplementary Fig. 4:

Same comments as Fig. 4 regarding the lengthy x-axis.

Supplementary Fig. 6:

Please list the numerical values of A_b and A_0 . Also, in the 3D field line plot, you are trying to make the point that the core is highly twisted, but the field lines seem to show increasing twist moving outward. Maybe this is just a characteristic of the particular field lines that are chosen, but as constructed, the plot is more confusing than helpful.

Replies to the Referees:

We again thank the three reviewers for constructive comments. We have made efforts to improve the manuscript by addressing the issues raised. Major changes made in the manuscript as shown in blue are as follows,

- 1. We have made a major restructuring of the text for better clarity.**
- 2. We have substantiated the arguments for the formation of the MFR in the corona without a significant preexistence by taking into account the formation timing and location of the MFR's feet, in addition to the flare morphology and evolution.**
- 3. We have fully taken into account the possibility that the filament may be associated with a flux rope.**

Below are the point-to-point replies to the comments, typed in boldface.

Reviewers' comments:

Reviewer #1 (Remarks to the Author):

2nd Review of "Formation of a Highly Twisted Magnetic Flux Rope During the Course of A Solar Eruption" by Wang et al.

The authors have responded satisfactorily to the comments and criticisms of the reviewers, including myself, added clarifications and corrections to the text, and made the figures easier to understand. In my opinion the paper can be accepted for publication. I have a few suggestions for corrections to fix grammatical or typographical errors, and a couple of comments intended for the authors only (no changes needed in the text).

- 1) The first sentence of the abstract should begin with "The"
- 2) "heliospheric" is misspelled in line 4 of the abstract (h and p are reversed)
- 3) "feet" should be "foot" in line 6 of the abstract
- 4) pg. 4, para. 3, line 3: as per my previous review, "slipping" should have "apparently" before it
- 5) pg. 5, para. 3, lines 5-7: the sentence beginning with "A 'seed'" is confusing. Here is a suggested rewrite (ignore italics): If a 'seed' MFR existed prior to the flare, it must have been too small to be resolved by AIA; otherwise the initial footpoint brightening would be ring-like rather than point-like.
- 6) pg. 6, para. 1, and pg. 7, last para.: "numeric" should be "numerical"
- 7) pg. 6, para. 3, line 4: replace "to the 3D occasion" by "to three dimensions"
- !!8) pg. 12, last line: the projection could be either toward or away from the observer. Either would reduce the projected speed.
- 9) pg. 13, para. 3: replace "a highly unlikely situation" by "under the unlikely assumption"
- 10) pg. 26, Table 1: "Shafranov" is misspelled.

Reply: Thanks for pointing out these grammatical/typographical errors, we have corrected them in the text.

11) In the response to my review, the authors remark that, in their opinion, the debate on pre-eruption configuration mostly originates from ambiguity in observations. I would point out that there are significant differences among models in terms of relative timings of key phases (e.g., flare reconnection vs CME fast acceleration) and underlying magnetic field (e.g., breakout requires a multipolar configuration, whereas the ideal instability and tether cutting models do not). On the other hand, all models agree that a flux rope exists after the eruption starts, so post-eruption signatures are not that useful for testing/constraining the models.

Reply: Thanks for the comments. We think the twist profile could be useful in discriminating the models.

Reviewer #2 (Remarks to the Author):

The paper was not modified enough to change my opinion. I still think that the authors' interpretation of results is not the only possible and is not substantiated enough. The title may mislead a reader that the paper presents the strong evidence of formation of the MFR exclusively during the eruption. However, it is only one of possibilities following from the observations.

Reply: We have substantiated our arguments for the dynamic formation of the MFR in the corona without a significant preexistence, from four different aspects (please see the subsection titled "Dynamic MFR Formation"). We do believe that we have provided the best plausible interpretation of the data.

Although we cannot exclude completely the possibility of a preexisting flux rope without direct measurements of coronal field, a problem that cannot be solved in the foreseeable future, we do have strong evidence against a preexisting flux rope anchored at the same place as the eruptive one, due to the unprecedented morphological evolution of flare ribbons. We have revised the title as "Buildup of a highly twisted magnetic flux rope during a solar eruption" to fully account for the possible existence of a "seed" for the eruptive flux rope (please see Discussion and Supplementary Information).

I cannot agree that the initial flux-rope footpoint brightening should be ring-like rather than point-like. Every brightening starts from separate points and then spreads into threads, ribbons, etc.

Reply: The unprecedented morphological evolution revealed in the current observation is that the footpoint brightening originates from the ends of flare ribbons and then expands during the flare impulsive phase in all directions into a closed bright ring, within which coronal dimming is developed. This is very different from brightening starting from separate points and then spreading into ribbons. We have further clarified this in the text.

Dense filament material does not necessarily fill the whole length of a flux rope or other magnetic structure supporting it. It is usual that the ends of an eruptive filament are rather far from previously visible ends of the undisturbed filament. The filament in NOAA 12443 evolved significantly before the

eruption. About 07 UT, four hours before the eruption, there were two close filaments within the same filament channel with different endpoints, as it is clearly visible in H-alpha images of Kanzelhoehe Observatory. At late phases of an eruption, many filaments show magnetic relation to remote areas of the solar surface. It can be the manifestation of coronal reconnection or activation of coronal loops (e.g., Grechnev et al., 2008, Solar Phys. 253, 263; van Driel-Gesztelyi et al., 2014, ApJ 788, 85). Anyway, these processes do not reveal the formation of flux ropes.

The authors easily threw off my pointing to the evidence of the presence of the MFR before the eruption, while they insist on the presence of a not more evident sheared arcade.

Reply: We agree that filament material may occupy part of a filament channel, and a filament may extend or become fragmented along a filament channel, which is always aligned along the polarity inversion line, but a filament in equilibrium is never seen to deviate far away from PIL to our best knowledge. The filament in our case experienced various moderate activities prior to the eruption, but its two feet were relatively fixed, and were further highlighted by a hot, apparently twisted loop in 131A (see Supplementary Information). In contrast, the conjugate dimming regions are located deeply in the flux concentrations far away from the PIL. The dimming regions have nothing to do with "remote reconnections" observed in some filament eruptions, which are shown as brightening with filament material draining back to the surface (e.g., Wang et al. 2017, ApJ, 834, 38, and references therein).

We have studied more carefully the evolution of the filament and indeed found some signatures that it may be associated with a flux rope of moderate twist (~1 turn; see Supplementary Information), but this does not compromise our argument that the eruptive MFR is formed dynamically during the eruption without a significant preexistence. Please see the evidence and reasoning given in the subsection titled "Dynamic MFR Formation", and discussion on the distinction between the filament-associated MFR and the eruptive MFR.

The sheared arcade is clearly manifested as sheared coronal loops with respect to the polarity inversion line, and further confirmed by NLFFF modeling and MHD simulation.

The authors still consider the motion of a loop-like structure in EUV images as radial and make wrong correction of the projected speed. They admit that the structure of the wavefront ahead of the erupting MFR is supposed to be dome-like and the loop appears in EUV images due to the effect of line-of-sight integration. In this case with the nearly radial direction of the line-of-sight, the optical density is maximum on a nearly vertical side of the dome but not at the top section. Thus the observed loop moves rather more transverse the line-of-sight than along it as supposed the authors.

Reply: We have removed the part on the correction of projected speeds.

Reviewer #3 (Remarks to the Author):

First, given the authors' response to my previous request to differentiate this work from the previous results of Webb et al., I am willing to grant that the detailed analysis of the magnetic fluxes in the

corona and at 1 AU constitutes a substantial advance in the analysis of a dynamically formed flux rope and its associated magnetic cloud. That being said, for the reasons I outline below, I find that a major revision is still required in order for this paper to be suitable for publication.

After repeated readings of this manuscript, the following structure is my best understanding of the authors' intent for this work:

1. The flux rope forms dynamically in the corona:
 - a. Footpoints are defined by bright rings that start as point-like brightenings
 - b. Subsequent dimmings within the bright footpoint rings indicate mass drainage
 - c. No detection of a coherent flux rope before the eruption (NLFFF, DARE)
 - d. Minimal net footpoint current when mapped to the pre-eruption Jz
2. The flux rope has a non-uniform twist profile with a highly twisted core:
 - a. Time-dependent flux accounting in the corona (brightened pixel tracking)
 - b. In situ observations of the resulting magnetic cloud (ACE, WIND)
 - c. Total flux comparison between corona and 1 AU (multiple methods)
 - d. Twist profile comparison between corona and 1 AU (GS reconstruction)

(I understand there may be a third conclusion, which is that the magnetic cloud has more flux than the observed coronal rope, but the authors need to determine if there is enough evidence to warrant treating this as a stand-alone conclusion or whether to fold it into the analysis of the twist profile.)

In the above structure, there are two key results, each of which is supported by four 'observations and analysis' bullet points. Frankly, the present layout of the paper does a poor job of conveying these results to the reader. In fact, many of the salient bullet points listed above are buried in the methods section and barely referenced in the main body. Conversely, the main body contains some ancillary information that doesn't help the reader understand the key results. While this ancillary information (such as the details of the magnetic cloud charge states) is important, it can be moved to the methods section in order to clear the way for more information in the main body that directly pertains the key results. Given these considerations, I am requesting that the authors undertake a major structural revision of this manuscript in order to make the key results more accessible to the reader.

Reply: We are very grateful to the careful reading and constructive comments, and have made a major structural revision as suggested.

Abstract:

The authors assume that the reader implicitly understands that "in situ spacecraft measurements" means measurements acquired at 1 AU (L1). This is not obvious to the general reader and should be stated in such a way that the two are explicitly connected.

Starting with "Here we identify..." every sentence should be aimed at the two key results of the paper. Something like, "Here we identify the dynamic formation of a flux rope during a solar eruptive event..."

The sentence beginning with "These results suggest..." tells the reader that the results in the previous sentences support the formation of a MFR with a highly twisted core. But the previous sentence only talks about the bright footpoint rings and the dimmings. The results that support the formation of a MFR with a highly twisted core are the flux accounting measurements, which are not directly mentioned here. They need to be mentioned in the abstract in order for the conclusions to make sense.

Finally, the abstract does not give enough credence to the analysis of the magnetic cloud at 1 AU. A full sentence in the abstract should be dedicated to the 1 AU analysis (GS solution, twist profile).

Reply: We have tried our best to revise the abstract as suggested, within the 150 word limit.

Introduction:

In paragraph 2, I refute the authors' assertion that solar eruptions are believed to be governed by a single physical process. As the authors themselves state in their reply to the reviewers, a given event could be influenced by breakout reconnection, the torus instability, or a combination of the two. As such, multiple physical processes can clearly be involved.

Reply: The different mechanisms are often too closely coupled to be clearly separated in an eruptive process. For clarification, we rephrased the sentence as follows, ".....solar eruptions, which are manifested as diversely as coronal mass ejections (CMEs), solar flares, and prominence eruptions, but can be governed by similar physical mechanisms.

The last paragraph before the results should be expanded to give a more definitive outline of the paper (see below). Also, 'in situ diagnostics' should be something more like 'in situ measurements of the resulting magnetic cloud at 1 AU.'

Reply: We have revised the last paragraph as suggested.

Results section:

I recommend a restructuring of the subheadings in this section as described below.

Present structure:

- > Observations and Analysis
- > Conclusions and Discussion

Suggested new structure:

- > Observations of the 4 Nov 2015 eruptive event (corona + 1 AU)
- > Dynamic flux rope formation (corona)

- > Non-uniform twist profile (corona + 1 AU)
- > Discussion

The 'Observations' subheading should be a condensed version of the first few 'Results' paragraphs, and could also include a paragraph on the in situ observations at 1 AU. Ancillary information like the details of the different AIA channels and the charge state observations of the magnetic cloud can be mentioned, but the bulk of the content should be moved to the Methods in order to condense this section and clear the way for more results (see below).

The next two subheadings should focus on conveying the 1a-1d and 2a-2d bullet points from above. For example, under the 'Dynamic formation' subheading, bullet point 1d on minimal net footpoint current should be described with a direct reference to the relevant data-driven modeling in the Methods section. At present, this result, which is one of the most important arguments against the existence of a pre-eruption flux rope, is not even mentioned in the main body of the paper.

As presently constructed, the main body does not do justice to the conclusion that the flux rope is believed to have a non-uniform twist profile with a highly twisted core. The entirety of this description is in the paragraph that begins with "The MFR is considered to be formally formed..." This description should be expanded and should include more information on the comparison between the $\Delta\Phi_p / \Delta\Phi_t$ coronal measurements and the twist profile in the GS reconstruction of the magnetic cloud. This a central result in the paper that should be appropriately emphasized.

Reply: We have restructured the Results section as suggested. Ancillary information is moved to the Methods or Supplementary Information, and the bullet points are explicitly emphasized.

Finally, substantial effort (and one of the six main figures) is devoted to analysis of the EUV shock wave and propagation speed of the eruption. Maybe I'm missing something, but this doesn't seem to be crucial to either of the key conclusions of the paper. Is it necessary to include this analysis? It seems like ancillary information at best. Consider moving it all to the Methods section.

Reply: The figure has been moved to Supplementary Information

A small note, but the plasma beta should be properly defined in the text.

Reply: Plasma beta has been defined in the text.

=====
Main figures:

Figure 3:

The virtual slits (S1-S4) could be better labeled so that the reader doesn't have to flip back and forth. For instance S3 -> S3 (FP-) and S4 -> S4 (FP+).

Reply: We have made the revision as suggested.

Figure 4:

Why does the x-axis cover such a long time range when all of the action is packed in the 13:30-14:30

time window? You could cut it off at 15:00 rather than 16:00 so that the reader can better examine the critical time window. Also, how much should we trust the early-time $\Delta \Phi_p / \Delta \Phi_t$ points that give the high central twist result? I know this is crucial to your conclusions, but it would seem to depend highly on how the footpoints (and therefore the toroidal flux) are defined early in time. As you state toward the end of the paper, caution is warranted when interpreting this result. I'll come back to this point in the discussion of the GS reconstruction of the MC in Figure 6.

Reply: We have cut the time axis off at 15:00 as suggested since most profiles are stabilized after 15:00. $\Delta \Phi_p / \Delta \Phi_t$ is most reliable during the flare impulsive phase, when both Φ_p and Φ_t increase with time rapidly. At earlier or later times, Φ_p and Φ_t change slowly and their ratio is more uncertain and noisy. Note $\Delta \Phi_p / \Delta \Phi_t$ is only meaningful when both terms are positive. We have given error bars in the revised figure, and taken two different approaches to calculate $\Delta \Phi_p / \Delta \Phi_t$ (see Methods). The time profiles are similar but one approach gives better result than the other, in comparison to in-situ diagnostics.

Figure 5:

As discussed above, the shock wave analysis could be moved to Methods since it doesn't add directly to the dynamic formation or non-uniform twist results. Does this contribute substantially enough to the dynamic flux rope formation result to warrant keeping as a main figure?

Reply: We have moved this figure to Supplementary Information as suggested

Figure 6:

This figure is barely addressed in the main text, but it is a key component of the non-uniform twist profile conclusion. With regard to the twist profile, the readout in Fig. 6c of the twist from the GS solution of the MC structure is key. Unfortunately, some important information is missing from the x-axis. In particular, there is no marking of the flux rope boundary, $|A_b - A_0|$. If I examine Supplementary Fig. 6 in detail, it looks like $A_0 \sim 155 \text{ T}\cdot\text{m}$ and $A_b \sim 40 \text{ T}\cdot\text{m}$. Does this mean that the x-axis cutoff in Fig. 6c at $|A - A_0| \sim 120 \text{ T}\cdot\text{m}$ is the flux rope boundary at $|A_b - A_0|$? As is common in the fusion community, one could consider plotting the normalized flux $|A - A_0| / |A_b - A_0|$, which would give a normalized x-axis running from 0 to 1.

Reply:

If $|A_b - A_0| \sim 120 \text{ T}\cdot\text{m}$, then the edge of the rope is as twisted as the core with a lower twist region in between. Why, then, do the $\Delta \Phi_p / \Delta \Phi_t$ observations in Fig. 4c not show a rise in $\Delta \Phi_p / \Delta \Phi_t$? It appears this latter injection phase is not tracked by the coronal measurements. The authors should flesh out these considerations for the reader.

Reply: While the GS reconstruction gives a definitive MC boundary resulting from the fitting of total pressure P_t as a function of the flux function A , it is impossible to infer the magnetic twist near the boundary of the flux rope under dynamic formation on the Sun. On the other hand, in the GS reconstruction the results reached by the four different approaches to derive twist diverge significantly toward the boundary, but converge toward the center. Hence, we should trust the twist profile close to the center more than toward the boundary.

I think the result that such high twist is observed in the flux rope without a kink instability has

interesting implications for the timescales. As the authors point out in their reply to the reviewers, the fast expansion of the highly twisted rope must nullify the impact of the kink instability. This could be stated more precisely in terms of the relevant timescales. The kink instability cannot arise faster than the Alfvén time along the rope. If the propagation itself is quasi-Alfvénic, then the kink instability won't have time to develop.

Reply: Thanks for the suggestion. In principle if a flux rope rises and expands faster than the development of the kink instability, then one will not see obvious writhing of the rope axis. By projecting the MC axis orientations derived from the models onto the solar surface (Supplementary Figure 7), we found that the MC rotated clockwise within 30 deg with respect to the MFR orientation on the Sun, which is approximated by the connection of its footpoints . The clockwise rotation is consistent with the MC's positive helicity sign as derived by the MC models. This has been clarified in Methods.

Methods section:

The NLFFF and DARE modeling should be explicitly referenced in the main body of the text in support of bullet points 1c and 1d.

Reply: The models have been explicitly referenced in the main text.

With regard to the projected speed of propagation, I find this subsection to still be confusing, and, as outlined above, I question its relevance to the paper. Consider revising further. For one, it's not clear to me where the new 1050 km/s lower bound comes from. There are half a dozen speeds listed in just a few sentences, and I can't tell what some of them mean and/or how they are related.

Reply: We have simplified this section.

Be sure to emphasize that the Bz mapping of ribbon and footpoint fluxes is derived from the pre-flare Bz map. This was not clear to me at first. I was confused about the zero net current comment until I realized it was zero net current before the flare.

Reply: We have emphasized the use of the preflare magnetogram in the text.

I can't seem to rationalize the flux accounting in Eq. 4. In particular, the factor of 2 in the $2\Phi_t$ term seems to result from circular logic. How is the total reconnected flux different from the total ribbon flux? It seems to me that the math is simpler: $\Phi_r = \Phi_R = \Phi_t + \Phi_p$ such that $\Phi_p = \Phi_R - \Phi_t$ (without the factor of two). Please try to clarify how the factor of two comes into the equation. Is there another reference that uses this flux accounting formulation? The factor of two clearly ripples through all of the interpretation of the poloidal flux, twist, flux ratios, etc. (e.g., Supplementary Table 1), so it is important to resolve this apparent contradiction.

Reply: The bright rings enclosing the MFR feet represent the footpoints of newly reconnected field lines belonging to the MFR, while the flare ribbons represent the footpoints of post-flare loops. Thus, the bright rings and the flare ribbons represent

footpoints of topologically distinct magnetic structures, though the former is only slightly dimmer than the latter. However, when we count the ribbon-swept flux (i.e., Φ_R), we have already taken Φ_t into account by counting the flux swept by the bright rings, because we cannot differentiate the bright rings from the flare ribbons due to their similar brightness (see Supplementary Video 4). Hence $\Phi_r = \Phi_R - \Phi_t$. We have further clarified this in the text.

List the values of A_0 and A_b from the GS reconstruction, in accord with Fig. 6 discussion.

Reply: We have listed the values of A_0 and A_b in the figure caption, and in the corresponding supplementary figure caption.

Supplementary figures:

Supplementary Fig. 4:
Same comments as Fig. 4 regarding the lengthy x-axis.

Reply: We have revised this figure as suggested.

Supplementary Fig. 6:
Please list the numerical values of A_b and A_0 . Also, in the 3D field line plot, you are trying to make the point that the core is highly twisted, but the field lines seem to show increasing twist moving outward. Maybe this is just a characteristic of the particular field lines that are chosen, but as constructed, the plot is more confusing than helpful.

Reply: The high twist toward the boundary results from field-line tracing, which gives different twist numbers than the other three approaches. We agree that this may cause some confusion and have removed the 3D field line plot. Numerical values of A_b and A_0 are now given in the figure caption of the P_t vs A plot.

REVIEWERS' COMMENTS:

Reviewer #3 (Remarks to the Author):

The referee thanks the authors for the substantial effort that was invested to improve the manuscript following the previous round of reviews. I have only minor suggestions to improve the clarity of the text in several places. Otherwise, I recommend the manuscript for publication.

=====
=====

Abstract:

I still think that more information about the in situ methods could be included in the final sentence. In order to clear space, the sentence beginning with, "The expansion begins..." could be eliminated entirely. It is likely too detailed to be understood by the reader in the abstract.

=====
=====

Introduction:

At the end of the first paragraph, I would say "...but may be governed..." rather than "...but can be governed..."

=====
=====

Overview of the 4 Nov 2015 Eruptive Event:

In the first paragraph, the authors refer to "an unprecedented morphology" with respect to this solar eruptive event. At face value, this implies that this morphology has not previously occurred on the Sun (a statement which is obviously unknowable). What they actually mean is that the observations of this morphology are unprecedented. Please consider revising. Note that this occurs again in the next section.

At the end of this section, the sentence beginning with, "Hence, an MFR must be present during the solar eruption..." is a bit awkward. Consider something like, "Hence, the solar eruption indisputably ejected an MFR into interplanetary space, regardless of whether that MFR is pre-existing or newly formed."

In the last sentence, consider a stronger transition such as, "Below we will explore in detail the evolution of the flare morphology and show that the MFR is in fact dynamically formed during the eruption."

=====
=====

Dynamic MFR formation:

Consider a stronger transition in the sentence beginning with, "We conclude that the observed MFR was dynamically formed..." Maybe something about using multiple observation and analysis techniques to reach this conclusion.

Under in-situ diagnostics: "Moreover, *the* average charge..."

The "unprecedented flare morphology and evolution" problem again crops up in the last paragraph of this section. Please correct.

=====
=====

Discussion:

The newly added final paragraph is a bit awkward. The words "remarkable" and "phenomenal" seem too strong and out of place. Consider, "It is also worth noting that the filament could play an important role..." and, "However, the post-eruption flux rope and the filament exhibit several important differences: (1) The filament..."

=====
=====

Methods:

In the paragraph beginning with, "We calculated the signed current..." there are two typos:

(1) "...no strong current density at either *foot*..."

(2) "...can be calculated by *integrating*..."

Also, "in the ballpark" is a bit informal. Consider "The results from both approaches are in rough agreement."

In the paragraph beginning with "The heating of *the* lower atmosphere during flares..." the word "deposit" should be replaced with "deposition" in two places.

=====
=====

Replies to the Reviewers:

We are grateful to the reviewer for the careful reading and constructive comments, and have made changes to the text as suggested. Below are the point-to-point replies to the comments, typed in boldface.

REVIEWERS' COMMENTS:

Reviewer #3 (Remarks to the Author):

The referee thanks the authors for the substantial effort that was invested to improve the manuscript following the previous round of reviews. I have only minor suggestions to improve the clarity of the text in several places. Otherwise, I recommend the manuscript for publication.

=====
=

Abstract:

I still think that more information about the in situ methods could be included in the final sentence. In order to clear space, the sentence beginning with, "The expansion begins..." could be eliminated entirely. It is likely too detailed to be understood by the reader in the abstract.

We have simplified the sentence about the timing of the expansion and included the information about the in-situ method, i.e., the Grad-Shafranov reconstruction.

=====
=

Introduction:

At the end of the first paragraph, I would say "...but may be governed..." rather than "...but can be governed..."

Done as suggested

=====

Overview of the 4 Nov 2015 Eruptive Event:

In the first paragraph, the authors refer to "an unprecedented morphology" with respect to this solar eruptive event. At face value, this implies that this morphology has not previously occurred on the Sun (a statement which is obviously unknowable). What they actually mean is that the observations of this morphology are unprecedented. Please consider revising. Note that this occurs again in the next section.

This is revised as "a morphology that has not been noticed before".

At the end of this section, the sentence beginning with, "Hence, an MFR must be present during the solar eruption..." is a bit awkward. Consider something like, "Hence, the solar eruption indisputably ejected an MFR into interplanetary space, regardless of whether that MFR is pre-existing or newly formed."

Done as suggested

In the last sentence, consider a stronger transition such as, "Below we will explore in detail the

evolution of the flare morphology and show that the MFR is in fact dynamically formed during the eruption."

Done as suggested

=====

=

Dynamic MFR formation:

Consider a stronger transition in the sentence beginning with, "We conclude that the observed MFR was dynamically formed..." Maybe something about using multiple observation and analysis techniques to reach this conclusion.

We have added "Based on the above observations and multiple analysis techniques".

Under in-situ diagnostics: "Moreover, *the* average charge..."

Corrected

The "unprecedented flare morphology and evolution" problem again crops up in the last paragraph of this section. Please correct.

We have replaced "unprecedented" with "newly discovered"

=====

=

Discussion:

The newly added final paragraph is a bit awkward. The words "remarkable" and "phenomenal" seem too strong and out of place. Consider, "It is also worth noting that the filament could play an important role..." and, "However, the post-eruption flux rope and the filament exhibit several important differences: (1) The filament..."

Done as suggested

=====

=

Methods:

In the paragraph beginning with, "We calculated the signed current...", there are two typos:

(1) "...no strong current density at either *foot* ..."

(2) "...can be calculated by *integrating* ..."

Also, "in the ballpark" is a bit informal. Consider "The results from both approaches are in rough agreement."

Corrected

In the paragraph beginning with "The heating of *the* lower atmosphere during flares...", the word "deposit" should be replaced with "deposition" in two places.

"deposit" can be used as a noun.

=====